# Taxometer: Improving taxonomic classification of metagenomics contigs

Svetlana Kutuzova[1,2,3], Mads Nielsen[1], Pau Piera[2,3], Jakob Nybo Nissen[2,3,5] ✉ & Simon Rasmussen[2,3,4,5] ✉

For taxonomy based classification of metagenomics assembled contigs, current methods use sequence similarity to identify their most likely taxonomy. However, in the related field of metagenomic binning, contigs are routinely clustered using information from both the contig sequences and their abundance. We introduce Taxometer, a neural network based method that improves the annotations and estimates the quality of any taxonomic classifier using contig abundance profiles and tetra-nucleotide frequencies. We apply Taxometer to five short-read CAMI2 datasets and find that it increases the average share of correct species-level contig annotations of the MMSeqs2 tool from 66.6% to 86.2%. Additionally, it reduce the share of wrong species-level annotations in the CAMI2 Rhizosphere dataset by an average of two-fold for Metabuli, Centrifuge, and Kraken2. Futhermore, we use Taxometer for benchmarking taxonomic classifiers on two complex long-read metagenomics data sets where ground truth is not known. Taxometer is available as open-source software and can enhance any taxonomic annotation of metagenomic contigs.

Metagenomic classifiers annotate reads or contigs with taxonomic information by searching for similar substrings in a collection of reference sequences. The quality of annotations depends on the chosen method[1–8] as well as the reference database, requiring a careful selection of tools depending on the research context. Due to the high complexity of metagenomics data and inevitable database incompleteness, full sample characterization is not usually achieved.

Most metagenomic classifiers rely on sequence similarity and work by querying databases for matches to individual contigs. If any individual contig has no match in the database, the string cannot be classified. However, in the related field of metagenomic binning, contigs from the same organism are successfully linked according to contig-contig feature similarity, such as tetra-nucleotide frequencies (TNFs) or abundances[9–14]. These links are used to group contigs to reconstruct metagenome assembled genomes (MAGs) from poorly studied environments with insufficient database representation,

without the use of a reference database[15–17]. Binning implies that a contig without a database match could be linked to a contig with the database match via similarity of feature vectors.

The taxonomic label is a hierarchical object containing a path from the highest taxonomic rank, phylum to a more precise label, such as species. A classification problem can be reduced to only predict a class on the selected taxonomic level (e.g. species) provided as a one-hot vector at the training time. However, a deep hierarchical loss widely used in e.g. the field of computer vision[18–20] reflects phylogenetic similarity of the true and the predicted node, and therefore allows modeling loss for the full lineage of nested labels. Similar hierarchical approaches for microbial data were previously used for predicting the taxonomies from different data types such as rRNA sequences[21,22] and Fourier-transform infrared spectroscopy[23,24]. To the best of our knowledge, such loss has not been applied in the field of DNA sequences taxonomic classification, where the prediction are either

[1]Department of Computer Science, University of Copenhagen, Universitetsparken 1, Copenhagen 2100, Denmark. [2]The Novo Nordisk Foundation Center for Protein Research, University of Copenhagen, Blegdamsvej 3A, Copenhagen 2200, Denmark. [3]The Novo Nordisk Foundation Center for Basic Metabolic Research, University of Copenhagen, Blegdamsvej 3A, Copenhagen 2200, Denmark. [4]The Novo Nordisk Foundation Center for Genomic Mechanisms of Disease, Broad Institute of MIT and Harvard, Cambridge 02142 MA, USA. [5]These authors contributed equally: Jakob Nybo Nissen, Simon Rasmussen. ✉e-mail: jakob.nissen@sund.ku.dk; srasmuss@sund.ku.dk

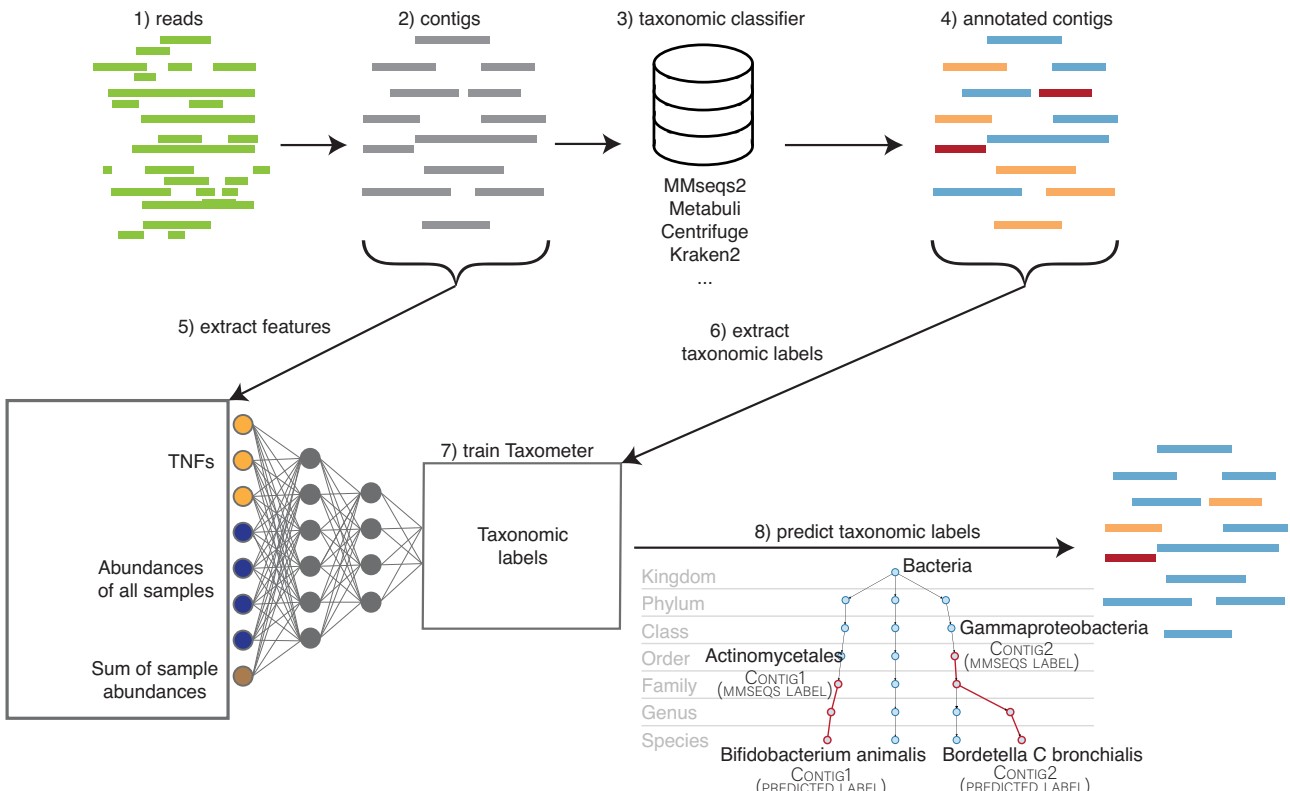

**Fig. 1 | Workflow.** Taxonomic profiling workflow using Taxometer. After assembling the metagenomic reads, contigs are annotated with any taxonomic classifier. Contigs are processed via Taxometer to extract abundances and tetra-nucleotide frequencies. Taxometer is then trained on the extracted features to predict the taxonomic annotations. A hierarchical loss allows training the model on the annotations from all levels. The predictions are then made for all taxonomic levels for each contig (e.g. completing the annotation from Actinomycetales order to Bifidobacteruim animalis species). For each label, a score is provided.

limited to one taxonomic level[25–28] or the functionality of the deep hierarchical loss is approximated by chaining neural network layers[29].

In this work we propose a method to enhance taxonomic profiling by utilizing the features that are used for binning. Our method, Taxometer, is designed to run on any standard metagenomics dataset of one or more samples where annotation of the contigs is required. Taxometer is based on a neural network that uses TNFs, abundances, and taxonomic labels from any metagenomic classifier. By the use of an abundance vector, Taxometer utilises the multisample experiment setup, that, to the best of our knowledge, was never attempted in the context of taxonomic classification before. The neural network is then trained to predict taxonomy for this particular dataset using the subset of contigs that have an annotation. Finally, the trained network is applied to the input contigs resulting in refined taxonomic labels of contigs as well as annotation of contigs without taxonomic labels (Fig. 1). As the taxonomic labels contain the full path from the taxonomic root to the placement on the bacterial taxonomic tree, we implemented a tree based hierarchical loss (Supplementary Fig. 1). This allows partial annotations not covering all taxonomic levels and that Taxometer can predict across all taxonomic levels accompanied by scores ranging from 0.5 to 1. After selecting the labels that pass the user-defined threshold, the workflow results in refined taxonomy annotations, containing more true positives and fewer false positives compared to the input taxonomy classification.

## Results

### Improving contig annotations of short-read based metagenomics

To demonstrate that Taxometer could improve the annotations of different taxonomic classifiers, we trained it on MMseqs2[3] and Metabuli[30] configured to use the GTDB database[31] and on Centrifuge[5]

and Kraken2[2] configured to use the NCBI database[32]. For the CAMI2 human microbiome datasets we found that MMseqs2 correctly annotated on average 66.6% of contigs at species level (Fig. 2a). When applying Taxometer, trained on the MMseqs2 annotations, the amount of correct annotated contigs increased to 86.2%. Additionally, when applied to two more challenging datasets, CAMI2 Marine and Rhizopshere, Taxometer increased the MMseqs2 annotation level from 78.6% to 90%, and from 61.1% to 80.9%, respectively (Fig. 2c). This was reflected in F1-score improvements of the annotation of between 0.1 and 0.13 for the human microbiome datasets. When we applied Metabuli, Centrifuge, and Kraken2 to the CAMI2 human microbiome datasets, they correctly annotated on average >94.8% of contigs at species level. Here, Taxometer did not improve on this close-to-perfect annotation level but also did not decrease performance (F1-score absolute change < 0.002). However, when applied to the Marine and Rhizosphere datasets, the performance of the three methods was much lower. For instance, Metabuli provided wrong species annotations for 12.7% and 37.6% of the two datasets, respectively (Fig. 2c, Supplementary Fig. 2). Here, Taxometer reduced the number of wrong annotations to 7.6% and 15.4%, increasing F1-score from 0.87 to 0.88 and from 0.61 to 0.69 respectively. Similarly, for Centrifuge and Kraken2 applied to the Rhizosphere dataset, Taxometer reduced the amount of wrong annotations from

68.7% to 39.5% (F1-score from 0.22 to 0.27) and from 28.7% to 13.3% (F1-score from 0.64 to 0.68), respectively (Supplementary Fig. 2). Even though Taxometer had high precision for taxonomic annotations it made mistakes and re-annotated a small amount of the correct MMseqs2 species-level annotations incorrectly (1.6%–3.3% on the CAMI2 human microbiome). We also performed the same analysis on the two mock communities: ZymoBIOMICS Microbial Community Standard with 10 strains and ZymoBIOMICS Gut Microbiome Standard

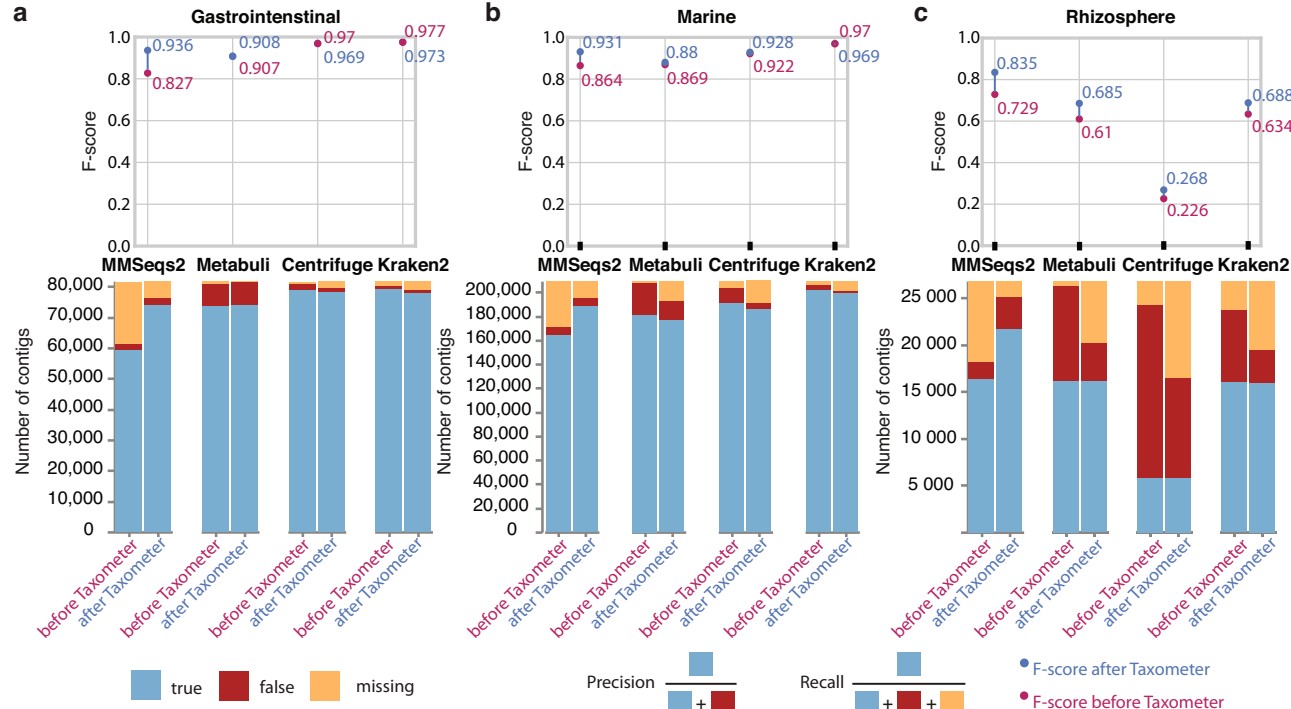

**Fig. 2 | CAMI2 results.** Taxonomic classifier annotations and Taxometer results at species level, compared to the ground truth, score threshold 0.95. MMseqs2 and Metabuli returned GTDB annotations, Centrifuge and Kraken2 returned NCBI annotations. The comparisons are to the ground truth labels. **(a)** CAMI2 Gastrointestinal, **(b)** CAMI2 Marine and **(c)** CAMI2 Rhizosphere datasets. Source data are provided as a Source Data file.

with 21 strains (Supplementary Fig. 3, Supplementary Fig. 4). Taxometer improved the quality of taxonomic annotations for both mock communities, except extremely well performing Kraken2 and MetaMaps, where the performance stayed the same after applying Taxometer (F1-score around 0.91). For MMseqs2, the F1-score improved from 0.28 to 0.847 for ZymoBIOMICS gut microbiome standard sample and from 0.623 to 0.889 for ZymoBIOMICS microbial community standard sample. Finally, we investigated the importance of varying the threshold of the annotation score. Here, we found that a threshold score of 0.95 provided a good balance between recall and precision for multi-sample datasets (Fig. 3, Supplementary Fig. 5, Supplementary Fig. 6). Taken together, we found that Taxometer could fill annotation gaps and remove incorrect taxonomic labels of large numbers of contigs from diverse environments, while only mislabeling a small minority of correctly labeled contigs.

## Using both abundance and TNF improved predictions

Given the ability of Taxometer to correctly predict new as well as correct wrong annotations we investigated the contribution of abundance and TNFs features for the predictions. We, therefore, trained Taxometer to predict MMseqs2 annotations for the CAMI2 human microbiome datasets using either the abundances or TNFs (Fig. 4a, Supplementary Fig. 7). Here we found that for higher taxonomic levels (phylum to genus) up to 98% of Taxometer annotations could be reproduced by training using only TNFs. This is in concordance with previous findings that TNFs could be used to classify metagenomics fragments at the genus level and that abundance showed better strain-level binning performance compared to TNFs[14,33]. The number of correct species labels predicted by the model that combined both TNFs and abundances was 18–35% larger than the models that only used TNFs or abundances for MMseqs2 annotations of the CAMI2 Airways dataset.

Since the abundance vector was an important feature for predicting the labels, we investigated if the annotations were still

improved if the abundance vector only consisted of one sample. We, therefore, only used the contigs from one sample from each of the 5 human microbiome CAMI2 datasets (Airways, Oral, Skin, Urogenital, Gastrointestinal). We observed that Taxometer still showed a major improvement for the MMseqs2 annotations (F1-score increased from 0.738 to 0.866 for the Airways dataset) and only slightly decreased the performance for the best performing classifiers (with the largest drop in F1-score for the Skin dataset from 0.926 to 0.895), supporting our previous findings when using the multi-sample abundance vector (Supplementary Fig. 8). Thus, using Taxometer was beneficial in both one-sample and multi-sample experiments.

Most metagenome binners use the abundance vector as well, so we benchmarked Taxometer against the VAMB binner as a taxonomic labels refinement tool. We ran VAMB on the CAMI2 datasets, which results in contigs classified to bins. For each bin we determined the assigned taxonomy by selecting the majority taxonomic label of its contigs using the Kraken2 taxonomic classifier. We assigned the bin taxonomic label to each contig in this bin. Comparing the assigned labels to ground truth, we determined that taxonomic classification after this procedure was worse than both Kraken2 classification and the Taxometer refinement of Kraken2 classification (86% correctly annotated contigs with the binning approach vs 91% from Kraken2 results) (Supplementary Fig. 9). Thus, binning alone cannot serve as a taxonomic refinement tool, despite its use of the contigs abundances.

## Novel species were predicted at genus level

To explore the limitations of Taxometer we investigated the performance when species in the dataset were missing from the database. To achieve this we deleted annotations from five species in the CAMI2 human microbiome MMseqs2 results before training Taxometer. This resulted in removing between 649 and 5127 contig annotations per dataset. As the deleted species were not in the training set, a perfect classifier should assign missing labels to the contigs that belong to this species. In our experiments, Taxometer predicted the correct genus

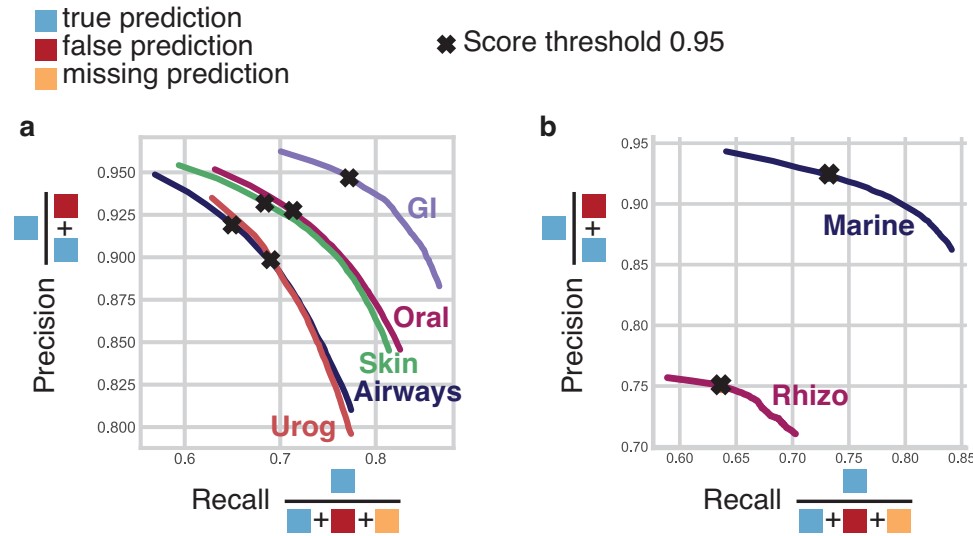

**Fig. 3 | Precision-recall curves.** Precision-recall curves for predictions at species level, compared to the ground truth, **(a)** CAMI2 human microbiome, **(b)** Marine and Rhizosphere datasets. Values given the score threshold 0.95 are marked with a cross sign. Source data are provided as a Source Data file.

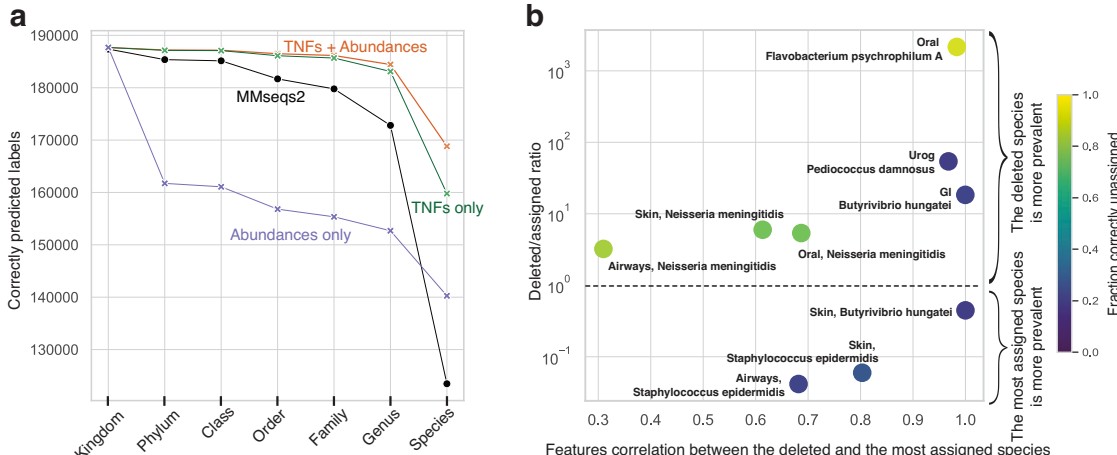

**Fig. 4 | Analysis of feature importance and novel taxa. a** Contribution of abundances and TNFs features to Taxometer performance demonstrated on the CAMI2 Airways short-read dataset. The amount of correctly predicted contigs labels at each taxonomic level using a score threshold of 0.5. **b** Simulation analysis of unknown taxa. X-axis: Pearson correlation coefficient between the mean feature vectors of the deleted and the assigned species. Y-axis: ratio between the number of contigs of the deleted species ("deleted") and the number of contigs of the species

that was the most prevalent among the incorrectly assigned ("assigned") in the training set. The color legend shows the share of correctly missing labels, equal to $1 - FP$, where FP is the share of false positives. FP is high when the assigned species was more prevalent in the training set and TNFs and abundances are highly correlated between the deleted and the assigned species. Source data are provided as a Source Data file.

label for all these contigs. However, the share of incorrectly assigned annotations at species level varied between 6% and 82% across the species and the datasets. For these wrong annotations, we found that the number of contigs of the deleted and the assigned species, and the mean feature correlation between them were the most important factors (Fig. 4b). We found that false positives tended to occur when the assigned species were more prevalent in the training set than the deleted species, e.g. in the Airways dataset *Staphylococcus aureus* was assigned to 1312 of 1879 contigs from the deleted species *Staphylococcus epidermidis*. That can be explained by *Staphylococcus epidermidis* being 17 times more prevalent in the training set than *Staphylococcus aureus*, with 78% of all contigs from the *Staphylococcus* genus were from *Staphylococcus aureus*. Second, we found it to make mistakes when TNFs and abundances were highly correlated between the deleted and the assigned species. For instance, in the Gastrointestinal dataset, 183 out of 229 contigs of the deleted *Butyrivibrio*

*hungatei* species were assigned to *Butyrivibrio sp900103635*, that only had 8 contigs in total. However, the pearson correlation of the mean feature vectors for these two species was 0.99. Thus, for the contigs of a novel taxon, Taxometer might assign a closely related taxon instead of returning a missing annotation.

## Taxometer improved recall for annotations on all confidence levels

Some taxonomic classifiers provide an interface to threshold the confidence of the taxonomic labels that they assign. As this will affect the precision of the resulting annotations we tested Taxometer performance for five different confidence level values of the Kraken2 classifier on the CAMI2 datasets (Supplementary Fig. 10). For instance, using the CAMI2 Gastrointestinal dataset and increasing the confidence of Kraken2 resulted in reduced F1-score 0.977 for the confidence value 0.0 to 0.859 for the confidence value 0.25, 0.825 for the

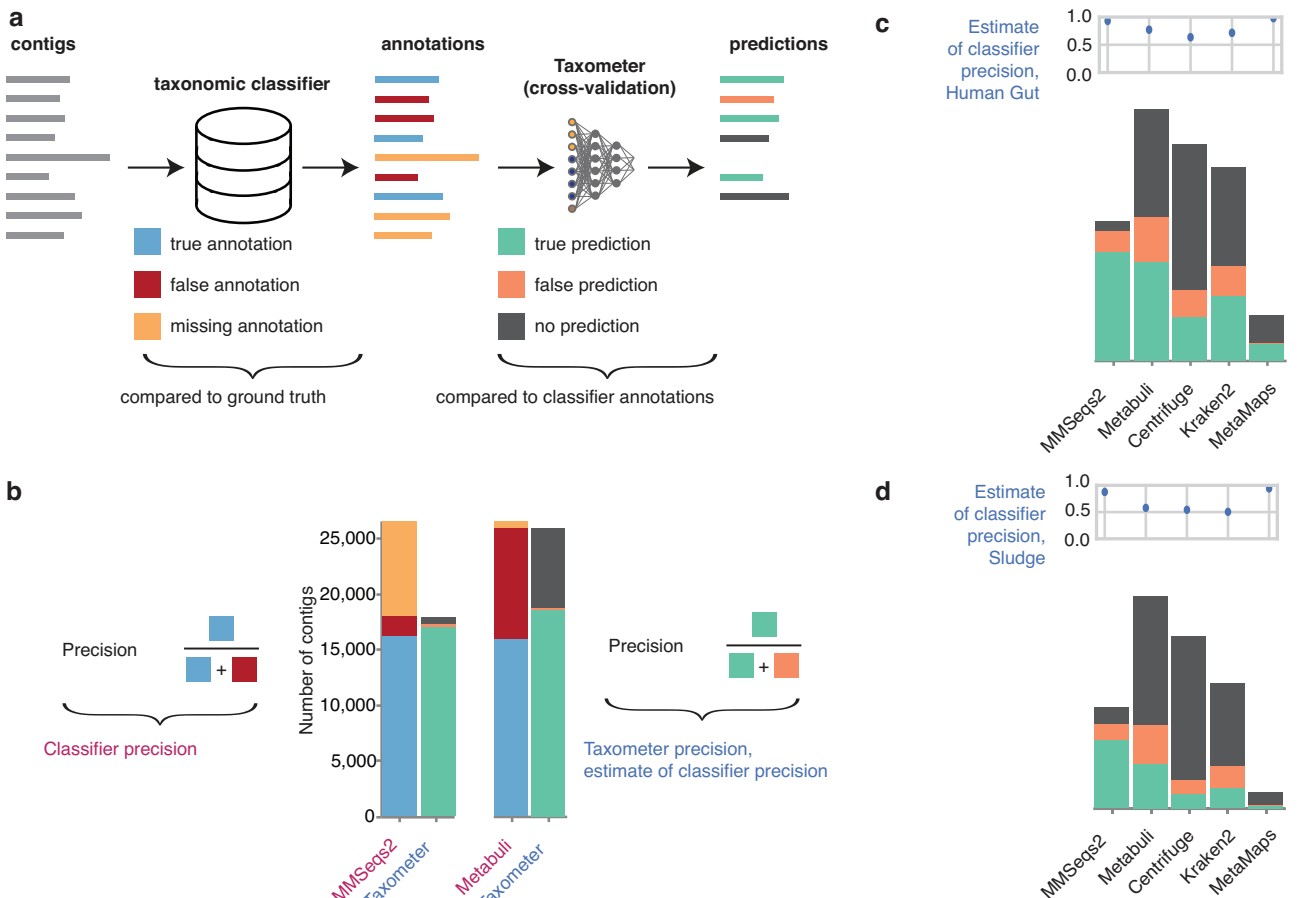

**Fig. 5 | Taxonomic profilers benchmarks and analysis of long-read datasets.**
**a** K-fold evaluation description. Taxometer predictions are compared to the classifer annotations, not the ground truth labels. **b** An example of the number of true positives, false positives, and false negatives used in the k-fold evaluation, species level for the Rhizosphere dataset, MMSeqs2 and Metabuli classifiers. The total number of contigs for Taxometer predictions equals the number of annotations initially returned by a classifier. **c**, **d** Real long-read datasets k-fold evaluation for Human Gut and Sludge datasets. Source data are provided as a Source Data file.

confidence value 0.5, 0.726 for the confidence value 0.75, and 0.029 for the confidence value 1.0 (Supplementary Fig. 11). For the Rhizosphere dataset, where Kraken2 showed low precision with the default value 0.0, increasing the confidence value from 0.0 to 0.25 increased F1-score from 0.715 to 0.717. Here, Taxometer improved F1-score for confidence level 0.25 to 0.731. While further increasing the confidence value to 0.5 and 0.75, Kraken2 F1-score dropped to 0.49 and 0.327, but Taxometer prediction stayed almost the same, with F1-score 0.72 for both confidence levels (Supplementary Fig. 12). In conclusion, filtering the output of a classifier by confidence level will result in better precision but lower recall. Here, we showed that Taxometer could improve recall and precision leading to a better F1-score when applied to the filtered output.

**Taxometer as a benchmarking tool**
We were interested in evaluating the performance of Taxometer for the annotation of long-read based metagenomics. In absence of a sufficiently large dataset with ground truth available, we analysed ZymoBIOMICS Gut Microbiome Standard with 21 strains and one sample. Despite an abundance vector only consisting of a single number, Taxometer improved the F1-score of MMSeqs2 classifier from 0.28 to 0.854, and made a 0.0-0.05 improvement for other classifiers. However, this dataset did not reflect real world data size and complexity. We, therefore, investigated if the consistency of Taxometer annotations could be used as a measure of classifier performance when ground truth labels were missing. Because metagenomics binning has been able to generate hundreds of thousands of MAGs, TNFs and

abundances carry a strong signal for contigs of the same origin[17]. Thus, we hypothesized that the ability of Taxometer to predict classifier annotations could reflect the performance of the classifier. Specifically, the more incorrect labels a classifier assigns to contigs of the same origin, the harder it will be for Taxometer to reproduce the labels assigned by a classifier. Therefore, inconsistent annotations by a classifier will decrease the score that Taxometer assigns to any taxa in the dataset. Thus, Taxometer scores can be used as a proxy for classifier consistency.

To investigate this we acquired annotations of four classifiers for the seven CAMI2 datasets, ZymoBIOMICS Microbial Community Standard and ZymoBIOMICS Gut Microbiome Standard, as well as the additional MetaMaps classifier for ZymoBIOMICS Gut Microbiome Standard, resulting in 37 sets of taxonomic labels. We divided each dataset into five folds and trained Taxometer to predict the annotations of each classifier five times using a new fold as the validation set. We then compared how well Taxometer predictions corresponded to the annotations of the particular classifier and the ground truth (Fig. 5a, c, Supplementary Fig. 13, Table 1). We found that Taxometer and classifier precision metrics were correlated with Spearman Correlation Coefficient of 1.0 across 3 out of 9 datasets, reflecting perfect ranking, and >0.78 for 8 out of 9 datasets, which is equivalent to maximum of only one misranked classifier (Fig. 5b, Supplementary Fig. 14, Table 1). We also noticed that the dataset for which the Taxometer precision ranking was mostly incorrect (negative Spearman correlation) was a Microbial community dataset with only 303 contigs and only 1 sample. Thus, in the absence of ground truth labels the

**Table 1 | Spearman correlations between classifiers precisions evaluated on ground truth and Taxometer precisions evaluated with cross-validation**

| Dataset | Spearman correlation | Number of contigs | Number of classifiers |
|---|---|---|---|
| CAMI2 Airways | 0.8 | 177,654 | 4 |
| CAMI2 Skin | 1.0 | 166,513 | 4 |
| CAMI2 Oral | 0.8 | 200,587 | 4 |
| CAMI2 Gastrointestinal | 0.8 | 81,238 | 4 |
| CAMI2 Urogenital | 1.0 | 57,233 | 4 |
| CAMI2 Rhizosphere | 0.8 | 26,787 | 4 |
| CAMI2 Marine | 1.0 | 209,385 | 4 |
| ZymoBIOMICS Microbial Community Standard | −0.1 | 303 | 4 |
| ZymoBIOMICS Gut Microbiome Standard | 0.78 | 84 | 5 |

"Number of classifiers" column shows how many taxonomic classifiers were benchmarked (4 for short-read datasets, 5 for a long-read dataset).

precision of Taxometer prediction for classifier labels could be used as a benchmark for different taxonomic classifiers within a dataset.

### Benchmarking contig annotations of long-read datasets

We then identified two PacBio HiFi read datasets of complex metagenomes; 4 samples from the human gut microbiome and 3 samples from an anaerobic digestion reactor sludge[34]. These datasets did not have ground truth labels and when we applied either GTDB- or NCBI-based classifiers they disagreed for 28%–39% of the contigs at species level annotations (Supplementary Fig. 15). Therefore, we applied the k-fold evaluation scheme described above to use Taxometer precisions for benchmarking the classifiers. MetaMaps had the highest precision of 0.95, but also the lowest share of annotated contigs on the species level (16%, while MMseqs2 annotated 49%). MMseqs2 had the second highest precision, i.e. 0.84 for the human gut dataset (Metabuli 0.68, Centrifuge 0.62, Kraken2 0.68) (Fig. 5d, Supplementary Fig. 16a). The precision and recall values for the MMseqs2 classifier were within the range of values for the CAMI2 datasets ([0.8,0.95] for precision and [0.6,0.85] for recall) (Supplementary Fig. 16b). This was consistent with the performance of the classifiers on the most difficult CAMI2 dataset, Rhizosphere, and thus we concluded that MMseqs2 returned the most precise annotations for both the human gut microbiome and sludge datasets compared to any of the other classifiers.

## Discussion

In summary, Taxometer can improve taxonomic annotations of any contig-level metagenomic classifier. Importantly, as contig abundances are specific to a certain dataset Taxometer is trained on the fly on each dataset. Therefore, even though Taxometer is designed as a supervised deep learning method we do not transfer the model, try to generalize across datasets or pre-train on a different dataset.

Tested on the annotations of the two taxonomic classifiers across several real and simulated datasets, Taxometer both filled annotation gaps and deleted incorrect labels. Furthermore, Taxometer is designed as a lightweight tool that is less compute intensive than the taxonomic annotations themselves. For instance, annotation of the CAMI2 and long-read datasets with MMSeqs2 took 2–4 h, while training Taxometer with a single GPU took <30 min for all datasets (Supplementary Fig. 17). Similarly, training could be done using CPUs in 10–120 min.

Additionally, Taxometer provides a metric for evaluating the quality of annotations in the absence of ground truth. The analysis of the long-read datasets suggested that the performance was stable across sequencing technologies. The main limitation of applying Taxometer to a metagenomics dataset is that read-level data is needed

to calculate contig abundances for a particular dataset. Taxometer therefore cannot be used to annotate contigs downloaded from a large database without such information. However, taxonomic annotations can be improved in any metagenomics experiment where reads are available. This could be a metagenomics experiment run in-house in a lab or where corresponding read-level data is available for download from SRA or ENA. Therefore, we believe that Taxometer is broadly applicable for improving annotations across almost all metagenomics datasets.

While evaluating Taxometer on short-read datasets, we discovered that for 6 out 7 CAMI2 datasets (Airways, Oral, Skin, Urogenital, Gastrointestinal, and Marine) the benchmarked taxonomic classifiers performed with a close to perfect accuracy making it difficult to compare the performance. While CAMI2 datasets remain the field standard in evaluating metagenomic binners and taxonomic classifiers, it highlights the need for harder synthetic datasets, that better represent the real data. Evaluating Taxometer on the long-read data, we faced the opposite problem of not having a corresponding synthetic dataset with taxonomic annotations of contigs. Therefore, it was not possible to perform the same analysis as on the short-read datasets, relying on k-fold evaluation and Taxometer benchmarking capabilities for taxonomic classifiers. The number of new methods emerging in the fields of taxonomy annotation and metagenome binning call for new synthetic datasets, both for short- and long-read technologies, of the size and complexity of real-world data.

Even though Taxometer uses the abundance vector similar to the metagenome binners such as VAMB, simply assigning all the contigs in a bin with a majority taxonomic label makes the annotations worse, highlighting the contribution of the Taxometer network and hierarchical loss.

Taxometer utilizes a flat softmax hierarchical loss to take into account the structure of taxonomic labels. While hierarchical losses are far from new in the deep learning field, their application to taxonomic labels in the fields of metagenome binning and taxonomic classification remains sparse. Future work could investigate different hierarchical losses in deep learning based metagenomic tools potentially improving metagenomic binning and taxonomic classification. Furthermore, Taxometer uses TNFs to analyse the DNA information of a contig with a multilayer perceptron. This feature engineering approach can potentially be replaced by a more general DNA foundation model. In this case, instead of the extracted TNFs, a whole contig or its embedding, produced by a foundation model, will be the input to Taxometer. Future work in this direction will include benchmarking different foundation models and evaluating their ability to produce meaningful contigs embedding in metagenome data context. Combining such a model pretrained on a large amount of relevant genomes with a deep hierarchical loss showcased in this study is a promising avenue for the future of taxonomic classification methods.

The novel application of binning features used for classification of DNA sequences with a deep hierarchical loss paves the way for further advances in deep learning applied to metagenomic data. Taxometer can be applied to the results of any metagenome classifier using any bacterial phylogeny, making it applicable in a broad range of fields and bioinformatics workflows.

## Methods

### Evaluation datasets

For the short-read benchmarks, we used synthetic CAMI2 datasets from five human microbiomes (Airways, Oral, Skin, Urogenital, and Gastrointestinal) and two environmental microbiomes (Marine and Rhizosphere)[35]. For the CAMI2 evaluations, we compared the results of taxonomic classifiers and Taxometer to the provided ground truth labels for each contig.

We benchmarked the two ZymoBIOMICS Microbial Community Standards: ZymoBIOMICS Microbial Community Standard with 8

bacteria and 2 yeasts sequenced with a short-read technology, and ZymoBIOMICS Gut Microbiome Standard with 21 strains sequenced with a long-read technology. To acquire the analog of ground truth, we mapped the contigs to the provided reference genomes with BLAST, and the highest scoring result for each contig was used as the ground truth. For the analysis to be compatible with the GTDB database, we only used the contigs that mapped to bacterial genomes, resulting in 303 and 84 contigs correspondingly.

We also benchmarked Taxometer using two real PacBio HiFi long-read datasets, a 'human gut' dataset from human stool samples[34] and a 'sludge' dataset from anaerobic digestion reactor sludge (ENA accessions ERR10905741-ERR10905743). In the absence of a synthetic long-read dataset with ground truth labels, the evaluations were done on real datasets via k-fold evaluation.

## Data preprocessing
For short-read benchmarking, we used sample-specific assemblies for the seven CAMI2 datasets: Airways (10 samples), Oral (10 samples), Skin (10 samples), Gastrointestinal (10 samples), Urogenital (9 samples), Marine (10 samples), Rhizosphere (21 samples), and the Zymo-BIOMICS Microbial Community Standard with 1 sample. For each dataset we aligned the synthetic short paired-end reads from each sample using bwa-mem (v.0.7.15)[36] to the concatenation of per-sample contigs from the particular dataset. BAM files were sorted using samtools (v.1.14)[37]. For all datasets, we used only contigs ≥ 2000 base pairs (bp). For long-read benchmarking we used ZymoBIOMICS Gut Microbiome Standard with 1 sample, a human gut microbiome dataset with 4 samples and a dataset from anaerobic digester sludge with 3 samples[38], both sequenced using Pacific Biosciences HiFi technology. We assembled each sample using metaMDBG (v. b55df39)[39], mapped reads using minimap2 (v.2.24)[40] with the 'ax map-hifi' setting, and from there proceeded as with the short reads.

## Abundances and TNFs
Computation of abundances and TNFs was done using the VAMB metagenome binning tool[14]. To determine TNFs, tetramer frequencies of non-ambiguous bases were calculated for each contig, projected into a 103-dimensional orthonormal space and normalized by z-scaling each tetranucleotide across the contigs. To determine the abundances of each sample, we used pycoverm (v.0.6.0)[41]. The abundances were first normalized within sample by total number of mapped reads, then across samples to sum to 1. To determine sample-relative abundance, the sum of abundances for a contig was taken before the normalization across samples. The dimensionality of the feature table was then $N_c \times (103 + N_s + 1)$ where $N_c$ was the number of contigs, $N_s$ was the number of samples.

## Ground truth labels
For the five CAMI2 human microbiome datasets, the provided genomes sequences were classified with GTDB-tk[42] (v.2.4.0) tool which resulted in all the contigs annotated to species level with GTDB[31] identifiers. Ground truth labels for Marine and Rhizosphere datasets were converted from NCBI[43] to GTDB (v.207) using the *gtdb to taxdump* tool (v.0.1.9) available at https://github.com/nick-youngblut/gtdb_to_taxdump (commit hash 24b82d6). Not all NCBI labels had a 1-to-1 match in GTDB and not all contigs were annotated in the CAMI2 dataset to the species level. The resulting number of fully annotated contigs used in the analysis was 208,783 out of 438,686 contigs (48%) for the Marine dataset and 26,734 out of 300,222 contigs (9%) for the Rhizosphere dataset. We approximated the ground truth taxonomic labels available for the ZymoBIOMICS Microbial Community Standards in the following way. The contigs of each dataset were mapped to the provided reference genomes using NCBI-BLAST (v2.15.0). For each contig, the hit with the highest bit-score was used as the ground truth label.

## Taxonomic classifiers
We obtained the taxonomic annotations for contigs of all seven short-read and two long-read datasets from MMseqs2 (v.7e2840)[3], Metabuli (v.1.0.1)[30], Centrifuge (v1.0.4)[5] and Kraken2 (v2.1.3)[2]. For the long-read dataset we also acquired the annotations with MetaMaps (v.633d2e)[44]. For MMseqs2, we used the *mmseqs taxonomy* command. For Metabuli, we used the *metabuli classify* command with *−seq-mode 1* flag. For Centrifuge, we used the *centrifuge* command with *-k 1* flag. For Kraken2, we used the *kraken* command with *−minimum-hit-groups 3* flag. For Kraken2 confidence level experiments we ran the the *kraken* command with *−-confidence-level* flags 0.0, 0.25, 0.5, 0.75 and 1.0. MMseqs2 and Metabuli were configured to use GTDB v207 as the reference database. Centrifuge, Kraken2 and MetaMaps were configured to use NCBI identifiers.

## Network architecture and hyperparameters
The input vector of dimensionality $N_c \times (103 + N_s + 1)$, described in "Abundances and TNFs" subsection, was passed through 4 fully connected layers (($(103 + N_s + 1) \times 512$, $512 \times 512$, $512 \times 512$, $512 \times 512$) with leaky ReLU activation function (negative slope 0.01), each using batch normalization (epsilon $1e - 05$, momentum 0.1) and dropout ($P = 0.2$). The output layer had dimensionality $512 \times N_l$ where $N_l$ was the number of leaves in the taxonomic tree (see subsection 4). The output is a vector of dimensionality $N_l$, where $N_l$ is the number of species in the taxonomic tree. The output entries are logits that undergo softmax transformation during the calculation of the hierarchical loss function, which is described in the next subsection. For all datasets, the network was trained for 100 epochs with batch size 1024 using the Adam optimizer with learning rates set via D-Adaptation[45]. The model was implemented using PyTorch (v.1.13.1)[46], and CUDA (v.11.7.99) was used when running on a V100 GPU.

## Hierarchical loss
A taxonomic tree was constructed for each dataset from the taxonomy classifier annotations for the set of contigs. Thus, the resulting taxonomy tree $T$ was a subgraph of a full taxonomy and the space of possible predictions was restricted to the taxonomic identities that appeared in the search results. We used a flat softmax loss, computed in the following way. Let $N_l$ be the number of leaves in the tree $T$. The network outputs the logit vector of dimensionality $N_l$, where each logit represents the value for each leaf of the taxonomic tree. The likelihoods of leaf nodes of the taxonomy tree were obtained from the softmax over the network output vector. The internal nodes of the tree are not the part of the output, so the likelihood of an internal node was then a sum of likelihoods of its children and computed recursively bottom-up. The complete model output was a vector of likelihoods for each possible label, including the internal nodes. For the backpropagation the negative log-likelihood loss was computed for all the ancestors of the true node and the true node itself. For example, if the true labels was at the genus level, its likelihood is equal the sum of likelihood of all the species that belong to this genus, and only the species logits are explicitly the part of the network output vector. Predictions were made for all taxonomic levels. For each level, starting from the root domain level, the node descendant with the highest likelihood was selected and returned as the part of the model's output in a resulting CSV file. If no node descendant had likelihood >0.5, the predictions from this level and the levels below were not included in the output.

## K-fold evaluation
All the contigs in a dataset that were annotated by a classifier are randomly divided into five folds. Taxometer was then trained five times, each time using one fold as a validation set and the remaining four folds as training set. The predictions were made for the five validation sets after training the network on the remaining four. The five

validation sets were then concatenated to again form the full dataset. This way the prediction for each contig was made without the prior knowledge of the classifier annotation of this contig. The F1-score metric that was used to benchmark the classifiers within a dataset was then calculated as following:

$$precision = \frac{TP}{TP + FP} \qquad (1)$$

$$recall = \frac{TP}{TP + FP + FN} \qquad (2)$$

$$F1\_score = \frac{2 \cdot precision \cdot recall}{precision + recall} \qquad (3)$$

where *TP* (true positives) is the total number of contigs for which the Taxometer prediction is the same as the classifier annotation, *FP* (false positives) is the total number of contigs for which the Taxometer prediction is different from the classifier annotation and *FN* (false negatives) is when Taxometer prediction is missing, but the classifier annotation exists. For this evaluation, a score threshold of 0.5 was used.

### Reporting summary
Further information on research design is available in the Nature Portfolio Reporting Summary linked to this article.

## Data availability
TheCAMI2 datasets were downloaded from https://data.cami-challenge.org/participate from "2nd CAMI Toy Human Microbiome Project Dataset" (5 human microbiome datasets), "2nd CAMI Challenge Marine Dataset" (Marine), "2nd CAMI Challenge Rhizosphere challenge" (Rhizosphere). The long-read human gut dataset is available at https://downloads.pacbcloud.com/public/dataset/Sequel-IIe-202104/metagenomics/. The long-read sludge dataset is available at the ENA as part of the study PRJEB39861. The ZymoBIOMICS datasets are available from https://zymoresearch.eu/collections/zymobiomics-microbial-community-standards/products/zymobiomics-gut-microbiome-standard and https://zymoresearch.eu/collections/zymobiomics-microbial-community-standards/products/zymobiomics-microbial-community-standard. All data generated in this study are available as Source Data. Source data are provided with this paper.

## Code availability
All code can be found on GitHub at https://github.com/RasmussenLab/vamb and is freely available under the permissive MIT license[47].

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

## Acknowledgements

S.K., M.N., and S.R. were supported by the Novo Nordisk Foundation (grant NNF19SA0059348). P.P., J.N.N., and S.R. were supported by the Novo Nordisk Foundation (grant NNF20OC0062223). S.K., P.P., J.N.N., and S.R. were supported by the Novo Nordisk Foundation (grants NNF23SA0084103 and NNF14CC0001). S.R. was supported by the Novo Nordisk Foundation (grant NNF21SA0072102).

## Author contributions

S.K, J.N.N and S.R. designed the experiments. P.P. and J.N.N. pre-processed the datasets. S.K. wrote the software and performed the analysis. M.N., P.P., J.N.N. and S.R. provided guidance and input for the analysis. S.K. wrote the manuscript with contributions from all coauthors. All authors read and approved the final version of the manuscript.

## Competing interests

S.R. is the founder and owner of BioAI. The remaining authors declare no competing interests.
