## [Peer Review File · Nature Communications]

Taxometer: Improving taxonomic classification of metagenomics contigsREVIEWER COMMENTS

Reviewer #1 (Remarks to the Author):

This paper discusses the development of a new algorithm to classify contigs building upon many of the advances from binning algorithms. Overall, the algorithm looks useful and claims to have a substantial improvement in accuracy compared to tools such as Kraken.

Major comments

- There needs to be a little work on the validation. CAMI is great, but more benchmarks on synthetic data (not simulation-based) is warranted. Specifically, mock communities would be recommended to better assess the false positive rate (i.e. for 8 member mock communities, you shouldn't expect to have hits to hundreds of strains as many of the existing tools output).
- More baseline methods. Kraken is a reasonable baseline, but there are other tools that classify contigs such as JAMS : https://github.com/johnmcculloch/JAMS_BW . JAMS in particular has also observed strong improvement over Kraken, so it is probably a more serious competitor.
- It would help strength the manuscript to clarify why long-context transformers / RNNs could not be used to classify contigs. I'd think LinearAttention would also provide strong improvement over Kraken, and would be much faster than the proposed method, since it wouldn't need the additional mapping step.

Minor comments

- If the method could also be used to improve binning, it would help to clarify that. It would be interesting to see how this would compare against SemiBin2.
- Model pre-training -- is the model trained on the fly, or is it fixed and merely performing inference? This is also worth clarifying, since pre-training can be quite expensive, but could lead to much stronger accuracy and reduced runtime for taxonomic prediction
- Absolute abundances - this is a typo. Since there are no qPCR data, spike-ins, flow cytometry or other absolute quantification methods leveraged here, these data aren't absolute. Thus need to replace this with relative abundances.

Reviewer #1 (Remarks on code availability):

I did not access the code, but it looks sufficiently documented

Reviewer #2 (Remarks to the Author):

The authors present us a novel tool Taxometer, a neural network based method that improves taxonomic assignment at species rank by contig abundance per-sample, contig abundance across samples and tetra-nucleotide frequencies per-contig. The method uses information from a dataset as a whole to provide high resolution for taxonomic assignment to train a neural network for each dataset, compared to classic taxonomic classification tool that only uses information of a single contig. The method shows significant improvement to classification of MMseqs2 taxonomy. The authors also demonstrate the potential of using Taxometer as a benchmark tool when ground truth labels are not available. However, some concerns have been raised as I read through the manuscript.

1. Balance between overconfident and overconservative assignment has always been an open-challenge for taxonomic classifiers. Based on Figure 1 and the result in Figure 2 It seems to be the case that the behaviors of the classifier would greatly impact the performance of Taxometer. If the labels used for training is conservative and assigned at higher taxonomic rank, such as using MMseq2 taxonomy as classifier, Taxometer could take advantage of the dataset's complete profile and correctly push higher rank assignments into species rank. However, if the classifier is aggressive, then the model is only able to remove labels from contigs, instead correcting mislabeled contigs. What's the

cause of such a phenomenon? Is it possible that the user could generate a better training set by partitioning labels by the confidence level of the classification result (eg. for low confidence assignments, only include higher rank labels in the training set)?

2. The question is also related to the above question. Widely used taxonomic classification tools often provide users with parameters that can alter the behavior of the classification, such as confidence scoring in Kraken2 (<https://github.com/DerrickWood/kraken2/wiki/Manual#confidence-scoring>), --min-score and --min-sp-score in Metabuli. Centrifuge also reports hitLength for users to remove low confidence assignments. Could users achieve similar results to Taxometer by filtering the classification results by the confidence level? If so, that narrows the application spectrum of the method.

3. I'm not 100% convinced that Taxometer could be used as a benchmark tool for classifiers with datasets of unknown ground truth. Taxometer F-score is calculated based on Taxometer precision and Taxometer recall. How to interpret Taxometer F-score, as well as Taxometer precision and Taxometer recall? Is the score only for ranking purposes or can it be interpreted quantitatively? The ranking for Marine dataset using Taxometer F-score does not match ground truth F-score(Supp Figure S5). In what case, or what kind of datasets is the ranking reliable, since the Pearson correlation varies between different datasets?

4. While the algorithm and architecture of the method is novel to metagenomic classification task, the use of sequence similarity and abundance information to re-assign reads/contigs has been explored by the community. It would be nice to add one extra classifier that uses the similar idea into benchmark comparison. Metamaps, for example, (<https://www.nature.com/articles/s41467-019-10934-2>), uses EM algorithm for read assignment and distribution.

5. Please provide testing data and expected output for code testing purposes.

Minor fixes:

Figure 2a is labeled as human gut but referenced as human microbiome datasets in the result section, or CAMI2 Gastrointestinal in the figure legend, which is confusing.

Figure 5 labels for b, c are swapped, thus not matching figure description text.

In summary, Taxometer is a great complementary tool for improving classification for MMSeg2 taxonomy. The methodology is sound. However, some questions need answers before it can be convinced of its universal application.

Reviewer #2 (Remarks on code availability):

Installation instruction is easy to understand and complete without problem.

Testing was not complete. The repo does not provide complete testing inputs for the software, also missing expected output.

Reviewer #3 (Remarks to the Author):

In this manuscript, Kutuzova et al. present Taxometer, an approach to refine contig taxonomic assignments using a pseudo-binning approach. The idea is to use a machine learning approach derived from VAMB, a binning method from the same group, to process a set of contig classifications

The idea is interesting, but, in my mind, the authors have not made a convincing case for its value in practice over the obvious alternative of binning (using, for example, VAMB). Unless the authors can demonstrate a clearer benefit (#1 below) in real data (#3 below), I feel this manuscript is of limited interest. I write this as someone who does find this topic very interesting and, in fact, wish for more details on the machine learning aspects (see #2 and #8 below), but I would not recommend it to colleagues who are primarily motivated by practical applications.

MAJOR COMMENTS

1. There is no direct comparison with a binning approach. A simple approach is to first bin the contigs (using, for example, VAMB from the same authors) and then taxonomically classify each MAG using all its contigs. The disadvantage of this approach is that some contigs are not binned, but it would be important to quantify this effect, particularly because the contigs that are not binned are likely to be the same ones that Taxometer would not be able to refine.

2. While I realize that the target audience of this manuscript is not necessarily interested in the details of the machine learning, some minimal details would have been helpful. What is the actual output of the network? Fig 1 is too sparse (e.g., the network is a simple cartoon depiction, not an actual network structure).

3. Almost all of the evaluations are on simulated data (CAMI2). While I appreciate the difficulty in estimating the quality of predictions based on real data, the lack of strong real-data support contributes to my skepticism about the widespread utility of this approach.

4. There is no mention of whether multiple samples can be used to improve the results (VAMB introduced a very useful approach to use multiple samples).

MINOR COMMENTS

5. Fig 5 is very difficult to understand. There are several points that are individually minor, but they add up: I think the caption is switched between the two panels. In (c), there are two different shades of blue. In (d), what is the top F-score? There are two F-scores being used in the other panels.

6. Fig. 3 reuses the same colors for labeling both the biomes and the categories (true predictions, false predictions, etc.). This makes it harder to understand the figure.

7. L306: I think "phylogenetic tree" is incorrect. It should be "taxonomic tree" or "cladogram" (because there are no distances between the taxa, only the structure of the tree).

8. After reading the paragraph "Hierchical loss" a few times, I still do not know what the loss is. This is related to point #2 above: in the methods section, a clear explanation of the machine learning approach would have been helpful. Right now, it is describing how the loss is calculated even with details on what is output, but there is no sufficient detail so that one can understand what the function is. When the authors write "for the above experiments", is the implication that this is not what is used in the actual implementation?

Reviewer #3 (Remarks on code availability):

I did not run the code as I did not immediately have at hand a set of contigs, mappings, and annotations that I could use to test it. However, the code does appear to be following best practices, with automated test (including CI integration) and good documentation. I am also happy that the

authors have chosen to make this a module within VAMB rather than a stand-alone tool as I predict it will ensure long-term maintainability and availability (for example, there is already a bioconda package for VAMB).

We thank all the reviewers for their time and the thorough evaluation of our work. We found the comments to be insightful and helpful in better highlighting the benefits of the proposed method. We addressed the revisions and made the corresponding changes to the manuscript main text, the Methods section, the figures and the supplementary figures. Our point-by-point response is below.

Reviewer #1:

This paper discusses the development of a new algorithm to classify contigs building upon many of the advances from binning algorithms. Overall, the algorithm looks useful and claims to have a substantial improvement in accuracy compared to tools such as Kraken.

Major comments

- There needs to be a little work on the validation. CAMI is great, but more benchmarks on synthetic data (not simulation-based) is warranted. Specifically, mock communities would be recommended to better assess the false positive rate (i.e. for 8 member mock communities, you shouldn't expect to have hits to hundreds of strains as many of the existing tools output).

We thank the reviewer for the valuable feedback and appreciate the specific suggestion of the dataset. A mock community dataset indeed serves as the missing link between the fully labelled CAMI2 datasets and the real data with no ground truth available, which we agree was initially missing in our work.

We used the two ZymoBIOMICS Microbial Community Standards¹: ZymoBIOMICS Microbial Community Standard with 8 bacteria and 2 yeasts sequenced with a short-read technology, and ZymoBIOMICS Gut Microbiome Standard with 21 strains sequenced with a long-read technology. To acquire the analog of ground truth, we mapped the contigs to the provided reference genomes with BLAST, and the highest scoring result for each contig was used as the ground truth. For the analysis to be compatible with the GTDB database, we only used the contigs that mapped to bacterial genomes.

We obtained taxonomic classifications for both datasets with MMseqs2, Metabuli, Kraken2 and Centrifuge, and for the long-read Gut Microbiome Standard also with MetaMaps, and refined them with Taxometer, following the same workflows as in the previous experiments. Despite the fact that these datasets only have one sample, producing a less informative abundance vector, Taxometer improved the quality of taxonomic annotations for all datasets, except extremely well performing Kraken2 and MetaMaps, where the performance stayed the same after applying Taxometer (F1-score around 0.91). For MMseqs2, the F1-score improved from 0.28 to 0.847 for ZymoBIOMICS gut microbiome standard sample and from 0.623 to 0.889 for

¹ <https://zymoresearch.eu/collections/zymbiomics-microbial-community-standards>

ZymoBIOMICS microbial community standard sample. In conclusion, the previous findings are supported by this new validation.

We added the quantitative results of these experiments to Supplementary Figures S3 and S4 and Supplementary Data.

Changes to the manuscript text:

Lines 119-129: *“We also performed the same analysis on the two mock communities: ZymoBIOMICS Microbial Community Standard with 10 strains and ZymoBIOMICS Gut Microbiome Standard with 21 strains (Supplementary Figure S3, Supplementary Figure S4). Taxometer improved the quality of taxonomic annotations for the both mock communities, except extremely well performing Kraken2 and MetaMaps, where the performance stayed the same after applying Taxometer (F1-score around 0.91). For MMseqs2, the F1-score improved from 0.28 to 0.847 for ZymoBIOMICS gut microbiome standard sample and from 0.623 to 0.889 for ZymoBIOMICS microbial community standard sample. ”*

Lines 269-280: *“In absence of a sufficiently large dataset with ground truth available, we analysed ZymoBIOMICS Gut Microbiome Standard with 21 strains and one sample. Despite an abundance vector only consisting of a single number, Taxometer improved the F1-score of MMSeqs2 classifier from 0.28 to 0.854, and made a 0.0-0.05 improvement for other classifiers. However, this dataset did not reflect the real data size and complexity.”*

New Supplementary Figures:

Supplementary Figure S3. Benchmark of ZymoBIOMICS microbial community standard sample. **a** Taxonomic classifier annotations and Taxometer F-scores at species level, using results of BLAST to the reference genomes as gold standard. **b** The number of true, false and missing annotations for four taxonomic classifiers and predictions of Taxometer trained on each classifier, compared to the results of BLAST to the reference genomes. The score threshold value is 0.5.

Supplementary Figure S4. Benchmark of ZymoBIOMICS gut microbiome standard sample. a Taxonomic classifier annotations and Taxometer F-scores at species level, using results of BLAST to the reference genomes as gold standard. **b** The number of true, false and missing annotations for four taxonomic classifiers and predictions of Taxometer trained on each classifier, compared to the results of BLAST to the reference genomes. The score threshold value is 0.5

- More baseline methods. Kraken is a reasonable baseline, but there are other tools that classify contigs such as JAMS : https://github.com/johnmcculloch/JAMS_BW . JAMS in particular has also observed strong improvement over Kraken, so it is probably a more serious competitor.

We appreciate the suggestion of a classifier to highlight the refining capabilities of Taxometer. However, the official documentation says that the JAMS software can only be run on the Biowulf HPC system, which we do not have access to. The UNIX/Linux installation of JAMS software is mentioned as being a work in progress. We nevertheless made several attempts to install the proposed JAMS software on UNIX systems and Mac computers and did not succeed.

We agree with the reviewer that including more tools is beneficial for demonstrating the potential for broad use of Taxometer. We, as suggested by reviewer #2, included MetaMaps, a long-read taxonomic classifier. We would like to point out that Taxometer is not an independent taxonomic classifier, thus Taxometer is not to be benchmarked against the taxonomic classifiers, but only to be run using the annotation that were previously acquired using one.

MetaMaps showed strong performance comparable with Kraken2 on the new Gut Microbiome Standard dataset. In conclusion, both the MetaMaps and Kraken2 reached F-score of 0.9 on the Gut Microbiome Standard dataset, with little room for Taxometer to improve the annotations. Using Taxometer did not degrade the performance of the classifier. However, MetaMaps showed a very low recall on the real human gut and sludge long read datasets, only annotating fewer than 15% of the contigs for each dataset. To compare, the share of annotated contigs for MMseqs2 on species level is more than 45% for each dataset.

We changed the main text and added the quantitative results of this experiment to the Supplementary Figure S4 and S14 and the Supplementary Data.

Changes to the manuscript text:

Lines 301-304: *“To investigate this we acquired annotations of four classifiers for the seven CAMI2 datasets, ZymoBIOMICS Microbial Community Standard and ZymoBIOMICS Gut Microbiome Standard, as well as the additional MetaMaps classifier for ZymoBIOMICS Gut Microbiome Standard, resulting in 37 sets of taxonomic labels.”*

Lines 344-347: *“MetaMaps had the highest precision of 0.95, but also the lowest share of annotated contigs on the species level (16%, while MMseqs2 annotated 49%). MMseqs2 had the second highest precision, i.e. 0.84 for the human gut dataset (Metabuli 0.68, Centrifuge 0.62, Kraken2 0.68) (Figure 5d, Supplementary Figure S16a).”*

Changes to the main figures:

Figure 5. Taxonomic profilers benchmarks and analysis of long-read datasets.
a, K-fold evaluation description. The Taxometer predictions are compared to the classifier annotations, not the ground truth labels. **b**, An example of the number of true positives, false positives, and false negatives used in the k-fold evaluation, species level. The total number of contigs for Taxometer predictions equals the number of annotations initially returned by a classifier. **c**, **d**, Real long-read datasets k-fold evaluation for human gut and sludge environments.

New Supplementary Figures:

■ true ■ false ■ missing (compared to BLAST against the reference genomes)

Supplementary Figure S4. Gut community benchmarks. **a** Taxonomic classifier annotations and Taxometer F-scores at species level, using results of BLAST to the reference genomes as gold standard. **b** The number of true, false and missing annotations for four taxonomic classifiers and predictions of Taxometer trained on each classifier, compared to the results of BLAST to the reference genomes. The score threshold value is 0.5.

Supplementary Figure S16. K-fold results for the long-read datasets. a, The number of true, false and no predictions of Taxometer for four taxonomic classifiers, compared to classifiers annotations, long-read datasets (Human Gut, Sludge), all domains. **b**, Precision-recall curves for Taxometer scores in the range [0.5,1] for species labels, long-read datasets, MMseqs2 classifier.

- It would help strength the manuscript to clarify why long-context transformers / RNNs could not be used to classify contigs. I'd think LinearAttention would also provide strong improvement over Kraken, and would be much faster than the proposed method, since it wouldn't need the additional mapping step.

The reviewer raised an important point about the potential of LLMs and transformer-like architectures in DNA data analysis. Taxometer accepts two features as an input: contigs abundances and TNFs. Contigs abundances, which are in the form of a relatively short numerical vector of a length of the number of samples, can be straightforwardly processed with a multilayer perceptron. TNFs, on the other hand, are extracted from the contigs DNA sequences, and such feature engineering could in principle be replaced by a pre-trained DNA foundation model. Taxometer also introduces hierarchical loss functions in application to bacterial taxonomy, which, in combination with a DNA foundation model, can be a promising avenue for further taxonomic classification research.

The shortcomings of current DNA foundation models include: 1) the amount of data required for training; 2) long training time; 3) the overall embedding quality on a range of tasks which can more efficiently be solved by the traditional methods based on feature engineering.

In response to the reviewer's comment, to investigate the DNA foundation models in the context of contig taxonomic classification, we pre-trained a HyenaDNA (<https://arxiv.org/abs/2306.15794>) model with the 16000 bp context window on the GTDB database. We also did additional pre-training on the CAMI2 Urogenital dataset, after pre-training on GTDB. We used 1x A100 GPU with 40GB memory, with a total training time of around 10 days. We then computed the embeddings of a subset of the CAMI2 Urogenital dataset of 6000 contigs. These 6000 contigs were annotated with the ground truth species and strain identifiers. We varied the test size from 0.1 to 0.9 with a step of 0.1. We randomly assigned the corresponding share of 6000 contigs to be a test set, and trained a simple MLP classifier (`sklearn.neural_network.MLPClassifier`) on the remaining contigs. We compared the performance of three inputs (abundances, TNFs and embeddings) on predicting two labels (strain and species). The results are presented on the plot:

In conclusion, the large overhead of using the DNA foundation model did not pay off in this particular application (10 days of training HyenaDNA model vs under 3 minutes of training Taxometer). If we were to replace the simple TNF feature with a

costly embedding in the Taxometer method, the predictive power of the model would not significantly improve. Also, this experiment is only exploratory, and the effort and computational resources it would take to systematically benchmark different foundation models (e.g. LinearAttention vs RNNs/state-space models mentioned by the reviewer) would be better disseminated as the study of its own.

We agree with the reviewer that further investigations of DNA foundation models is an exciting research area and that the field of taxonomic classification might benefit from such models in the future. We modified the Discussion section to elaborate on this topic.

Changes to the manuscript text:

Lines 488-498: *“Furthermore, Taxometer uses TNFs to analyse the DNA information of a contig with a multilayer perceptron. This feature engineering approach can potentially be replaced by a more general DNA foundation model. In this case, instead of the extracted TNFs, a whole contig or its embedding, produced by a foundation model, will be the input to Taxometer. Future work in this direction will include benchmarking different foundation models and evaluating their ability to produce meaningful contigs embedding in metagenome data context. Combining such a model pretrained on a large amount of relevant genomes with a deep hierarchical loss showcased in this study is a promising avenue for the future of taxonomic classification methods.”*

Minor comments

- If the method could also be used to improve binning, it would help to clarify that. It would be interesting to see how this would compare against SemiBin2.

Taxometer can indeed be beneficial in the field of metagenome binning. However, showing its use in such an application requires a different neural network architecture and binning-specific benchmarks. We are currently working on a separate study, where we developed a metagenome binning model with the taxonomy as an additional input feature. Taxometer is used there as a part of the workflow. This new binner, called TaxVAMB, is already available from the master branch of the source code <https://github.com/RasmussenLab/vamb> while the manuscript is work in progress

- Model pre-training -- is the model trained on the fly, or is it fixed and merely performing inference? This is also worth clarifying, since pre-training can be quite expensive, but could lead to much stronger accuracy and reduced runtime for taxonomic prediction

The model is trained on the fly for each dataset. The reason for that is the use of contigs abundances which are dataset-specific. We updated the Methods and

Discussion sections in response to the reviewer's comments so that the explanation of the training procedure is more clear.

Changes to the manuscript text:

Lines 424-427: *"Importantly, as contig abundances are specific to a certain dataset Taxometer is trained on the fly on each dataset. Therefore, even though Taxometer is designed as a supervised deep learning method we do not transfer the model, try to generalize across datasets or pre-train on a different dataset."*

- Absolute abundances - this is a typo. Since there are no qPCR data, spike-ins, flow cytometry or other absolute quantification methods leveraged here, these data aren't absolute. Thus need to replace this with relative abundances.

We thank the reviewer for pointing that out. We are indeed dealing with relative abundances only. We corrected the wording, and now term this feature a sample-relative abundance. The difference between the abundance vector and the sample-relative abundance is that the sample-relative abundance is the sum of abundances for each contig across all samples.

We changed the incorrect wording in the main text and Figure 1.

Changes to the manuscript text:

Lines 541-542: *"To determine sample-relative abundance, the sum of abundances for a contig was taken before the normalization across samples"*

Reviewer #1 (Remarks on code availability):

I did not access the code, but it looks sufficiently documented

Thank you very much. In addition to code, we created example input and output for the easy testing of the tool. The instructions are by the link:

https://github.com/RasmussenLab/vamb/blob/taxometer_release/README_Taxometer.md#example-with-data

Reviewer #2:

The authors present us a novel tool Taxometer, a neural network based method that improves taxonomic assignment at species rank by contig abundance per-sample, contig abundance across samples and tetra-nucleotide frequencies per-contig. The method uses information from a dataset as a whole to provide high resolution for taxonomic assignment to train a neural network for each dataset, compared to classic taxonomic classification tool that only uses information of a single contig. The

method shows significant improvement to classification of MMseqs2 taxonomy. The authors also demonstrate the potential of using Taxometer as a benchmark tool when ground truth labels are not available. However, some concerns have been raised as I read through the manuscript.

1. Balance between overconfident and overconservative assignment has always been an open-challenge for taxonomic classifiers.

The reviewer raised a valuable question about the tradeoff between overconfident and overconservative assignments in taxonomic classifiers. As shown in our previous analysis, to solve this problem, Taxometer introduces a score assigned to each label on each taxonomic label, allowing the user more flexibility in making that decision, demonstrated on e.g. Figure 3.

Based on Figure 1 and the result in Figure 2 It seems to be the case that the behaviors of the classifier would greatly impact the performance of Taxometer. If the labels used for training is conservative and assigned at higher taxonomic rank, such as using MMseq2 taxonomy as classifier, Taxometer could take advantage of the dataset's complete profile and correctly push higher rank assignments into species rank. However, if the classifier is aggressive, then the model is only able to remove labels from contigs, instead correcting mislabeled contigs. What's the cause of such a phenomenon?

The reviewer correctly pointed out that the behaviour of the classifier on the different taxonomic levels will affect the Taxometer performance. In our original submission we compared the results on the Rhizosphere datasets of the MMSeqs2 classifier for MMseqs2 and Kraken2 (Supplementary Figure S2), where for MMseqs2 Taxometer was able to add the missing species labels correctly. For Kraken2, Taxometer was only able to remove incorrect species labels.

While on the species level MMSeqs2 and Kraken2 return a similar number of correct annotations, we observed that Kraken2 made more mistakes starting at the phylum level already. MMseqs2 returned more labels in total than Kraken2 on higher taxonomic levels (family and up), thus it can hardly be described as overconservative. We attribute the fact that Taxometer could only remove incorrect labels on the species level for Kraken2 and not predict them to the same extent as for the MMseqs2 classifier to the errors Kraken2 made on the higher taxonomic levels.

In conclusion, we attribute the Taxometer ability to predict correct labels not to the number of labels, misassigned by a classifier on the same taxonomic level (which is what usually described as the overconfident or overconservative behaviour), but to the number of correct assignments on higher taxonomic labels.

Is it possible that the user could generate a better training set by partitioning labels by the confidence level of the classification result (eg. for low confidence assignments, only include higher rank labels in the training set)?

As opposed to Taxometer, other classifiers (e.g. MMseqs2) only assign the confidence score to the whole taxonomic label, without an estimate on each taxonomic level. The only way to request a label on a higher taxonomic level with more confidence is to run the classifier again with a different confidence flag. We analyse this strategy in detail in the next paragraph.

2. The question is also related to the above question. Widely used taxonomic classification tools often provide users with parameters that can alter the behavior of the classification, such as confidence scoring in Kraken2 (<https://github.com/DerrickWood/kraken2/wiki/Manual#confidence-scoring>), --min-score and --min-sp-score in Metabuli. Centrifuge also reports hitLength for users to remove low confidence assignments. Could users achieve similar results to Taxometer by filtering the classification results by the confidence level? If so, that narrows the application spectrum of the method.

We thank the reviewer for this valuable suggestion of a follow-up experiment that takes the different confidence levels of a taxonomic classifier into account. We selected Kraken2 to run this experiment as: 1) one of the strongest classifiers on most of the CAMI2 datasets; 2) the classifier that provides a confidence flag that ranges from 0 to 1, making the selection of maximum and minimum values straightforward. We ran Kraken on all the 7 CAMI2 datasets with the following --confidence flag values: [0.0, 0.25, 0.5, 0.75, 1.0]. 0.0 is the default value, corresponding to our previous experiments displayed on the main figures. We present the quantitative results of this analysis on the Supplementary Figure S9. We observed that:

- For the human CAMI2 datasets (Airways, Oral, Skin, Gastrointestinal, Urogenital), changing the default value of the confidence flag from 0.0 to [0.25, 0.5, 0.75, 1.0] makes the Kraken2 output worse in quality, with a major drop in recall (F1-score drops from 0.977 to 0.859, 0.825, 0.726, and 0.029 for the Gastrointestinal dataset).
- For the difficult Rhizosphere dataset, where Kraken2 shows low F1-score with the default value 0.0 (F1-score 0.715), increasing the confidence value to 0.25 indeed resulted in increased precision and F1-score compared to the Kraken2 results with confidence value 0.0 (F1-score increased to 0.717). However, Taxometer was able to improve on that result as well, reaching higher recall and F-score than Kraken2 (improved F1-score to 0.731). Interesting that while further increasing the confidence value to 0.5 and 0.75 resulted in a major drop in recall for Kraken2, it did not affect the Taxometer corrections much. Taxometer produced almost the same quality labels for the confidence level 0.75, as it did for the confidence level 0.25, providing even higher overall gains. (F1-scores before Taxometer 0.717, 0.49 and 0.327, F1-scores after Taxometer 0.731, 0.72 and 0.72)

In conclusion, this experiment shows that filtering classifiers results by the confidence level can only result in better precision, but Taxometer can also improve recall and results in similar or better F-score when applied to the filtered classifier output. We added the quantitative results of this experiment to Supplementary Figures S10, S11 and S12 and Supplementary Data. Furthermore we updated the main text:

Changes to the manuscript text:

Lines 250-266: *“Some taxonomic classifiers provide an interface to threshold the confidence of the taxonomic labels that they assign. As this will affect the precision of the resulting annotations we tested Taxometer performance for five different confidence level values of the Kraken2 classifier on CAMI2 datasets (Supplementary Figure S10). For instance, using the CAMI2 Gastrointestinal dataset and increasing the confidence of Kraken2 resulted in reduced F1-score 0.977 for the confidence value 0.0 to 0.859 for the confidence value 0.25, 0.825 for the confidence value 0.5, 0.726 for the confidence value 0.75, and 0.029 for the confidence value 1.0 (Supplementary Figure S11). For the Rhizosphere dataset, where Kraken2 showed low precision with the default value 0.0, increasing the confidence value from 0.0 to 0.25 increased F1-score from 0.715 to 0.717. Here, Taxometer improved F1-score for confidence level 0.25 to 0.731. While further increasing the confidence value to 0.5 and 0.75, Kraken2 F1-score dropped to 0.49 and 0.327, but Taxometer prediction stayed almost the same, with F1-score 0.72 for both confidence levels (Supplementary Figure S12). In conclusion, filtering the output of a classifier by confidence level will result in better precision but lower recall. Here, we showed that Taxometer could improve recall and precision leading to a better F1-score when applied to the filtered output. ”*

New Supplementary Figures:

Supplementary Figure S10. Predictions at different confidence levels of Kraken2, CAMI2. True, false and missing labels returned by Kraken2 configured with different confidence levels and the corresponding Taxometer predictions.

3. I'm not 100% convinced that Taxometer could be used as a benchmark tool for classifiers with datasets of unknown ground truth. Taxometer F-score is calculated based on Taxometer precision and Taxometer recall. How to interpret Taxometer F-score, as well as Taxometer precision and Taxometer recall? Is the score only for ranking purposes or can it be interpreted quantitatively? The ranking for Marine

dataset using Taxometer F-score does not match ground truth F-score(Supp Figure S5). In what case, or what kind of datasets is the ranking reliable, since the Pearson correlation varies between different datasets?

We thank the reviewer for pointing our attention to the possible confusion about the use of Taxometer as the taxonomic classifiers benchmarking tool. Our assumption behind the idea of using Taxometer as a ranking tool is that the contigs with more similar abundance vectors are more likely to originate from the same genome. If the contigs from the same genome are annotated with the different labels, Taxometer will not have enough reference labels for this genome to make consistent predictions for all its contigs. Thus, the quality of Taxometer predictions reflects the quality of the classifier labels.

To make our message more clear, we replaced the F1-score metrics with the precision metrics, as this is likely the parameter that a user will be the most interested in. The recall can be trivially derived given the precision and it depends more on the dataset (how diverse and well studied the taxa are in this dataset) than on the classifier (how often the classifier is wrong given that it returned a label). Taxometer precision on the classifier's labels can then be interpreted as the measure of consistency of the labels within a subset of contigs that belong to the same genome.

Additionally, we replaced the quantitative evaluation using the Pearson correlation of all the data points with instead displaying quality of ranking using Spearman correlation per each dataset. The Spearman correlation is very strong (≥ 0.8) for 8 out of 9 analysed datasets, suggesting that only a maximum of 1 classifier per dataset was ranked incorrectly. We also notice that the dataset for which the Taxometer precision ranking was mostly incorrect (negative Spearman correlation) is a Microbial community dataset with only 303 contigs and only 1 sample, while the recommended dataset size starts with 10000 contigs and more than one sample. To conclude, the ranked Taxometer precisions for several classifiers within a dataset correlates strongly with the ranking of the classifiers for this dataset.

We described it in the main text and corrected the Figure 5 and the Supplementary Figure S8.

Changes to the manuscript text:

Lines 301-316: *“To investigate this we acquired annotations of four classifiers for the seven CAMI2 datasets, ZymoBIOMICS Microbial Community Standard and ZymoBIOMICS Gut Microbiome Standard, as well as the additional MetaMaps classifier for ZymoBIOMICS Gut Microbiome Standard, resulting in 37 sets of taxonomic labels. We divided each dataset into five folds and trained Taxometer to predict the annotations of each classifier five times using a new fold as the validation set. We then compared how well Taxometer predictions corresponded to the annotations of the particular classifier and the ground truth (Figure 5a,c,*

Supplementary Figure S13). We found that Taxometer and classifier precision metrics were correlated with Spearman Correlation Coefficient of 1.0 across 3 out of 9 datasets, reflecting perfect ranking, and more than 0.78 for 8 out of 9 datasets, which is equivalent to maximum of only one misranked classifier (Figure 5b, Supplementary Figure S14). We also noticed that the dataset for which the Taxometer precision ranking was mostly incorrect (negative Spearman correlation) was a Microbial community dataset with only 303 contigs and only 1 sample. Thus, in the absence of ground truth labels the precision of Taxometer prediction for classifier labels could be used as a benchmark for different taxonomic classifiers within a dataset.”

Changes to the main figures:

Figure 5. Taxonomic profilers benchmarks and analysis of long-read datasets. **a**, K-fold evaluation description. The Taxometer predictions are compared to the classifier annotations, not the ground truth labels. **b**, An example of the number of true positives, false positives, and false negatives used in the k-fold evaluation, species level. The total number of contigs for Taxometer predictions equals the number of annotations initially returned by a classifier. **c**, **d**, Real long-read datasets k-fold evaluation for human gut and sludge environments.

4. While the algorithm and architecture of the method is novel to metagenomic classification task, the use of sequence similarity and abundance information to reassign reads/contigs has been explored by the community. It would be nice to add one extra classifier that uses the similar idea into benchmark comparison.

Metamaps, for example, (<https://www.nature.com/articles/s41467-019-10934-2>), uses EM algorithm for read assignment and distribution.

We thank the reviewer for the suggestion of the classifier, highlighting the potential for broad use of our method. Similar to Taxometer, MetaMaps uses information from all the contigs in the dataset before making a decision about how to assign each one. We notice that MetaMaps does not use any of the multisample abundance information to refine the annotation. We would also like to point out that Taxometer is not an independent taxonomic classifier, thus Taxometer is not to be benchmarked against the taxonomic classifiers, but only to be run using the annotation that were previously acquired using one.

We added MetaMaps to our evaluations as another taxonomic classifier for the long-read datasets (Gut Microbiome Standard, Human Gut long-read dataset and Sludge long-read dataset) where it showed strong performance comparable with Kraken2 on the new Gut Microbiome Standard dataset. In conclusion, both MetaMaps and Kraken2 reached F-score of 0.9 on the Gut Microbiome Standard dataset, with little room for Taxometer to improve the annotations. Using Taxometer for that dataset did not degrade the performance of the classifier. However, MetaMaps showed a very low recall on the real human gut and sludge long read datasets, only annotating fewer than 16% of the contigs for each dataset. To compare, the share of annotated contigs for MMseqs2 on species level is more than 49% for each dataset.

We added the quantitative results of this experiment to the Supplementary Figure S4 and S16 and the Supplementary Data.

Changes to the manuscript text:

Lines 301-304: *“To investigate this we acquired annotations of four classifiers for the seven CAMI2 datasets, ZymoBIOMICS Microbial Community Standard and ZymoBIOMICS Gut Microbiome Standard, as well as the additional MetaMaps classifier for ZymoBIOMICS Gut Microbiome Standard, resulting in 37 sets of taxonomic labels.”*

Lines 344-347: *“MetaMaps had the highest precision of 0.95, but also the lowest share of annotated contigs on the species level (16%, while MMseqs2 annotated 49%). MMseqs2 had the second highest precision, i.e. 0.84 for the human gut dataset (Metabuli 0.68, Centrifuge 0.62, Kraken2 0.68) (Figure 5d, Supplementary Figure S16a).”*

Changes to the main figures:

Figure 5. Taxonomic profilers benchmarks and analysis of long-read datasets. **a**, K-fold evaluation description. Taxometer predictions were compared to the classifier annotations, not the ground truth labels. **b**, An example of the number of true positives, false positives, and false negatives used in the k-fold evaluation, species level. The total number of contigs for Taxometer predictions equals the number of annotations initially returned by a classifier. **c**, **d**, Real long-read datasets k-fold evaluation for human gut and sludge environments.

New Supplementary Figures:

■ true ■ false ■ missing (compared to BLAST against the reference genomes)

Supplementary Figure S4. Benchmark of ZymoBIOMICS gut microbiome standard sample. a Taxonomic classifier annotations and Taxometer F-scores at species level, using results of BLAST to the reference genomes as gold standard. **b** The number of true, false and missing annotations for four taxonomic classifiers and predictions of Taxometer trained on each classifier, compared to the results of BLAST to the reference genomes. The score threshold value is 0.5.

Supplementary Figure S16. K-fold results for the long-read datasets. a, The number of true, false and no predictions of Taxometer for four taxonomic classifiers, compared to classifiers annotations, long-read datasets, all domains. **b,** Precision-recall curves for Taxometer scores in the range [0.5,1] for species labels, long-read datasets, MMseqs2 classifier.

5. Please provide testing data and expected output for code testing purposes.

Thank you for the suggestion. The testing data and expected outputs are now a part of the code repository and their use is described in the README by the link

https://github.com/RasmussenLab/vamb/blob/taxometer_release/README_Taxometer.md#example-with-data

Minor fixes:

Figure 2a is labeled as human gut but referenced as human microbiome datasets in the result section, or CAMI2 Gastrointestinal in the figure legend, which is confusing.

Thank you, we have now revised Figure 2 to ensure consistent naming.

Figure 5 labels for b, c are swapped, thus not matching figure description text.

Again thank you, this has been corrected.

In summary, Taxometer is a great complementary tool for improving classification for MMSeq2 taxonomy. The methodology is sound. However, some questions need answers before it can be convinced of its universal application.

We thank the reviewer for insightful suggestions and questions. We implemented several new evaluations and revised the manuscript accordingly.

Reviewer #3

In this manuscript, Kutuzova et al. present Taxometer, an approach to refine contig taxonomic assignments using a pseudo-binning approach. The idea is to use a machine learning approach derived from VAMB, a binning method from the same group, to process a set of contig classifications

The idea is interesting, but, in my mind, the authors have not made a convincing case for its value in practice over the obvious alternative of binning (using, for example, VAMB). Unless the authors can demonstrate a clearer benefit (#1 below) in real data (#3 below), I feel this manuscript is of limited interest. I write this as someone who does find this topic very interesting and, in fact, wish for more details on the machine learning aspects (see #2 and #8 below), but I would not recommend it to colleagues who are primarily motivated by practical applications.

MAJOR COMMENTS

1. There is no direct comparison with a binning approach. A simple approach is to first bin the contigs (using, for example, VAMB from the same authors) and then taxonomically classify each MAG using all its contigs. The disadvantage of this approach is that some contigs are not binned, but it would be important to quantify this effect, particularly because the contigs that are not binned are likely to be the same ones that Taxometer would not be able to refine.

We thank the reviewer for suggesting this experiment that helps better highlight the focus of our method and its difference from binning. While Taxometer and VAMB, indeed, share the main input features, the hierarchical loss, first introduced in Taxometer, is what makes the taxonomic labels refinement possible. To support this statement we performed an additional experiment where we implemented the suggested approach in the following way: 1) we run VAMB on CAMI2 datasets, which results in contigs classified to bins; 2) for each bin we determine the assigned taxonomy by selecting the majority taxonomic label of its contigs (from the Kraken2 taxonomic classifier); 3) we assign the bin taxonomic label to each contig in this bin.

Comparing the assigned labels to ground truth, we determined that taxonomic classification after this procedure was worse than both Kraken2 classification and the Taxometer refinement of Kraken2 classification (86% correctly annotated contigs with the binning approach vs 91% from Kraken2 results). We conclude that binning with VAMB alone cannot serve as a taxonomic refinement tool, despite its use of the contigs abundances. We added this experiment to the main text and Supplementary Figure S9 and Supplementary Data.

Changes to the manuscript text:

Lines 202-212: *“Most metagenome bidders use the abundance vector as well, so we benchmarked Taxometer against the VAMB bidder as a taxonomic labels refinement tool. We run VAMB on CAMI2 datasets, which results in contigs classified to bins. For each bin we determined the assigned taxonomy by selecting the majority taxonomic label of its contigs using the Kraken2 taxonomic classifier. We assigned the bin taxonomic label to each contig in this bin. Comparing the assigned labels to ground truth, we determined that taxonomic classification after this procedure was worse than both Kraken2 classification and the Taxometer refinement of Kraken2 classification (86% correctly annotated contigs with the binning approach vs 91% from Kraken2 results) (Supplementary Figure S9). Thus, binning alone cannot serve as a taxonomic refinement tool, despite its use of the contigs abundances.”*

Lines 464-467: *“Even though Taxometer uses the abundance vector similar to the metagenome bidders such as VAMB, simply assigning all the contigs in a bin with a majority taxonomic label makes the annotations worse, highlighting the contribution of the Taxometer network and hierarchical loss.”*

New Supplementary Figures:

Supplementary Figure S9. VAMB bins for taxonomic refinement. The number of true, false and missing annotations for CAMI2 human microbiome datasets and Kraken2. Compared are: Kraken2 annotations, VAMB bin majority annotations, Taxometer refinement of Kraken2.

2. While I realize that the target audience of this manuscript is not necessarily interested in the details of the machine learning, some minimal details would have been helpful. What is the actual output of the network? Fig 1 is too sparse (e.g., the network is a simple cartoon depiction, not an actual network structure).

We thank the reviewer for pointing out that our explanation of the network architecture is insufficient. We added the details on the network output to the Methods section, alongside with the network architecture details in the subsection “Network architecture and hyperparameters”. We also revised the “Hierarchical loss” subsection and added Supplementary Figure S1 with the graphical explanation of the hierarchical loss.

Changes to the manuscript text:

Lines 579-590: “The input vector of dimensionality $N_c \times (103 + N_s + 1)$, described in “Abundances and TNFs” subsection, was passed through 4 fully connected layers ($(103+N_s+1) \times 512$, 512×512 , 512×512 , 512×512) with leaky ReLU activation function (negative slope 0.01), each using batch normalization (epsilon $1e-05$, momentum 0.1) and dropout ($P = 0.2$). The output layer had dimensionality $512 \times N_l$ where N_l was the number of leaves in the taxonomic tree (see subsection 4). The output is a vector of dimensionality N_l , where N_l is the number of species in the taxonomic tree. The output entries are logits that undergo softmax transformation during the calculation of the hierarchical loss function, which is described in the next subsection. For all datasets, the network was trained for 100 epochs with batch size 1024 using the Adam optimizer with learning rates set via D-Adaptation. The model was implemented using PyTorch (v.1.13.1)47, and CUDA (v.11.7.99) was used when running on a V100 GPU.”

New Supplementary Figures:

Supplementary Figure S1. Flat softmax hierarchical loss. The network output is a vector of N logits, where N is the number of leaves (species) on the taxonomic tree. The softmax likelihoods are recursively summed bottom up. The negative log likelihood is computed between the true and predicted labels on all taxonomic levels.

3. Almost all of the evaluations are on simulated data (CAMI2). While I appreciate the difficulty in estimating the quality of predictions based on real data, the lack of strong real-data support contributes to my skepticism about the widespread utility of this approach.

We strongly agree that CAMI2 datasets are not entirely reflective of the properties of real datasets. In response to this valid point, we added datasets that serve as the middle ground between the fully labelled CAMI2 datasets and the real data with no ground truth available.

As mentioned in our response to reviewer 1, we used the two ZymoBIOMICS Microbial Community Standards²: ZymoBIOMICS Microbial Community Standard with 8 bacteria and 2 yeasts sequenced with a short-read technology, and ZymoBIOMICS Gut Microbiome Standard with 21 strains sequenced with a long-read technology. To acquire the analog of ground truth, we mapped the contigs to the provided reference genomes with BLAST, and the highest scoring result for each contig was used as the ground truth. For the analysis to be compatible with the GTDB database, we only used the contigs that mapped to bacterial genomes.

We obtained taxonomic classifications for both datasets with MMseqs2, Metabuli, Kraken2 and Centrifuge, and for the long-read Gut Microbiome Standard also with MetaMaps, and refined them with Taxometer, following the same workflows as in the previous experiments. Despite the fact that these datasets only have one sample, producing a less informative abundance vector, Taxometer improved the quality of taxonomic annotations for all datasets, except extremely well performing Kraken2 and MetaMaps, where the performance stayed the same after applying Taxometer (F1-score around 0.91). For MMseqs2, the F1-score improved from 0.28 to 0.847 for ZymoBIOMICS gut microbiome standard sample and from 0.623 to 0.889 for ZymoBIOMICS microbial community standard sample. In conclusion, the previous findings are supported by this new validation.

We also validate our approach on real long-read datasets, from human gut and sludge environments. However, we are naturally limited in the kind of analysis we can perform since these datasets do not have the ground truth available.

We added the quantitative results of these experiments to Supplementary Figures S3 and S4 and Supplementary Data.

Changes to the manuscript text:

Lines 119-129: *“We also performed the same analysis on the two mock communities: ZymoBIOMICS Microbial Community Standard with 10 strains and ZymoBIOMICS Gut Microbiome Standard with 21 strains (Supplementary Figure S3, Supplementary Figure S4). Taxometer improved the quality of taxonomic annotations for the both mock communities, except extremely well performing Kraken2 and MetaMaps, where the performance stayed the same after applying Taxometer (F1-score around 0.91). For MMseqs2, the F1-score improved from 0.28 to 0.847 for ZymoBIOMICS gut microbiome standard sample and from 0.623 to 0.889 for ZymoBIOMICS microbial community standard sample. ”*

Lines 269-280: *“In absence of a sufficiently large dataset with ground truth available, we analysed ZymoBIOMICS Gut Microbiome Standard with 21 strains and one sample. Despite an abundance vector only consisting of a single number, Taxometer improved the F1-score of MMSeqs2 classifier from 0.28 to 0.854, and made a 0.0-*

² <https://zymoresearch.eu/collections/zymbiomics-microbial-community-standards>

0.05 improvement for other classifiers. However, this dataset did not reflect the real data size and complexity.”

New Supplementary Figures:

a

ZymoBIOMICS® Microbial Community Standard

b

Supplementary Figure S3. Benchmark of ZymoBIOMICS microbial community standard sample. a Taxonomic classifier annotations and Taxometer F-scores at species level, using results of BLAST to the reference genomes as gold standard. **b** The number of true, false and missing annotations for four taxonomic classifiers and predictions of Taxometer trained on each classifier, compared to the results of BLAST to the reference genomes. The score threshold value is 0.5.

Supplementary Figure S4 Gut community benchmarks a Taxonomic classifier annotations and Taxometer F-scores at species level, compared to the results of BLAST to the reference genomes. **b** The number of true, false and missing annotations for four taxonomic classifiers and predictions of Taxometer trained on each classifier, compared to the results of BLAST to the reference genomes. The score threshold value is 0.5

4. There is no mention of whether multiple samples can be used to improve the results (VAMB introduced a very useful approach to use multiple samples).

We thank the reviewer for pointing out the insufficient explanation of our method. Taxometer already uses multiple samples via contigs abundances as one of the

input features. The abundance vector is the main feature allowing to attend to the contigs' co-occurrence across the different samples. We changed the main text in response to this valuable correction.

Additionally, we were inspired by the reviewer's comment to perform a new experiment where we investigate the effect of the multi-sample vs one-sample setup.

Changes to the manuscript text:

Lines 76-78: *“By the use of an abundance vector, Taxometer utilises the multisample experiment setup, that, to the best of our knowledge, was never attempted in the context of taxonomic classification before.”*

Lines 191-201: *“Since the abundance vector was an important feature for predicting the labels, we investigated if the annotations were still improved if the abundance vector only consisted of one sample. We, therefore, only used the contigs from one sample from each of the 5 human microbiome CAMI2 datasets (Airways, Oral, Skin, Urogenital, Gastrointestinal). We observed that Taxometer still showed a major improvement for the MMseqs2 annotations (F1-score increased from 0.738 to 0.866 for the Airways dataset) and only slightly decreased the performance for the best performing classifiers (with the largest drop in F1-score for the Skin dataset from 0.926 to 0.895), supporting our previous findings when using the multi-sample abundance vector (Supplementary Figure S8). Thus, using Taxometer was beneficial in both one-sample and multi-sample experiments.”*

New Supplementary Figures:

Supplementary Figure S8. CAMI2 one-sample experiment. Performance of Taxometer on CAMI2 human microbiome datasets when only using contigs from one sample in each dataset. The score threshold value is 0.5.

MINOR COMMENTS

5. Fig 5 is very difficult to understand. There are several points that are individually minor, but they add up: I think the caption is switched between the two panels. In (c), there are two different shades of blue. In (d), what is the top F-score? There are two F-scores being used in the other panels.

We corrected the typos and inconsistencies in Figure 5, and edited the figure caption. We also removed one of the panels and changed the colour scheme to improve clarity.

Figure 5. Taxonomic profilers benchmarks and analysis of long-read datasets.
a, K-fold evaluation description. The Taxometer predictions are compared to the classifier annotations, not the ground truth labels. **b**, An example of the number of true positives, false positives, and false negatives used in the k-fold evaluation, species level for the Rhizosphere dataset, MMSeqs2 and Metabuli classifiers. The total number of contigs for Taxometer predictions equals the number of annotations initially returned by a classifier. **c**, **d**, Real long-read datasets k-fold evaluation for Human Gut and Sludge datasets.

6. Fig. 3 reuses the same colors for labeling both the biomes and the categories (true predictions, false predictions, etc.). This makes it harder to understand the figure.

Thank you for the suggestion. We corrected the colour scheme of Figure 3, so that visually distinct colours are used to mark the microbiomes and the metrics.

7. L306: I think "phylogenetic tree" is incorrect. It should be "taxonomic tree" or "cladogram" (because there are no distances between the taxa, only the structure of the tree).

Thank you, we corrected the wording in the main text and Methods.

8. After reading the paragraph "Hierarchical loss" a few times, I still do not know what the loss is. This is related to point #2 above: in the methods section, a clear

explanation of the machine learning approach would have been helpful. Right now, it is describing how the loss is calculated even with details on what is output, but there is no sufficient detail so that one can understand what the function is.

We thank the reviewer for pointing out that the explanation of the method is not sufficient. We substantially revised the “Hierarchical loss” subsection.

Changes to the manuscript text:

Lines 592-616: *“A taxonomic tree was constructed for each dataset from the taxonomy classifier annotations for the set of contigs. Thus, the resulting taxonomy tree T was a subgraph of a full taxonomy and the space of possible predictions was restricted to the taxonomic identities that appeared in the search results. We used a flat softmax loss, computed in the following way. Let N_l be the number of leaves in the tree T . The network outputs the logit vector of dimensionality N_l , where each logit represents the value for each leaf of the taxonomic tree. The likelihoods of leaf nodes of the taxonomy tree were obtained from the softmax over the network output vector. The internal nodes of the tree are not the part of the output, so the likelihood of an internal node was then a sum of likelihoods of its children and computed recursively bottom-up. The complete model output was a vector of likelihoods for each possible label, including the internal nodes. For the backpropagation the negative log-likelihood loss was computed for all the ancestors of the true node and the true node itself. For example, if the true labels was at the genus level, its likelihood equals the sum of likelihood of all the species that belong to this genus, and only the species logits are explicitly the part of the network output vector. Predictions were made for all taxonomic levels. For each level, starting from the root domain level, the node descendant with the highest likelihood was selected and returned as the part of the model’s output in a resulting CSV file. If no node descendant had likelihood > 0.5 , the predictions from this level and the levels below were not included in the output.”*

When the authors write “for the above experiments”, is the implication that this is not what is used in the actual implementation?

We corrected this misleading wording, since it is the implementation that was used for all the experiments. The text now reads: “We used a flat softmax loss...” instead of “For the above experiments, we used...”

Reviewer #3 (Remarks on code availability):

I did not run the code as I did not immediately have at hand a set of contigs, mappings, and annotations that I could use to test it. However, the code does appear to be following best practices, with automated test (including CI integration) and good documentation. I am also happy that the authors have chosen to make this a module within VAMB rather than a stand-alone tool as I predict it will ensure long-

term maintainability and availability (for example, there is already a bioconda package for VAMB).

We appreciate the reviewer's opinion on the matter of long-term software maintenance. This consideration was indeed behind our decision to include Taxometer as a part of VAMB.

For the convenience of the testing, we added the example data with additional instructions to the repository. The updated instructions can be found here https://github.com/RasmussenLab/vamb/blob/taxometer_release/README_Taxometer.md#example-with-data

REVIEWERS' COMMENTS

Reviewer #2 (Remarks to the Author):

The revised version of the manuscript adds extra datasets, tools to compare, and different parameter settings into the benchmarking analysis, which strength to significance and credibility of the manuscript. Adding synthetic datasets that are widely used for benchmarking studies into the analysis fills the gap between pure simulation and real world metagenomic data. The authors also added multiple methods that leverage the usage of abundance vector such as VAMB binner and MetaMaps, and the results suggests that Taxometer performs better overall at species rank annotations.

In addition, the authors replaced Taxometer F1-score with Taxometer precision to evaluate whether the software can be used as a benchmarking tool for the other classifiers. The idea behind using Taxometer as a benchmark tool is to measure the consistency between labels of contigs originated the same taxon, which is sound to me.

The authors also provided a detailed explanation of the algorithm and architecture of the machine learning model used in this method, which makes the method more clear and easier to understand.

Overall, I'm satisfied with the revision of the manuscript. The work supports the conclusions and claims. The software was tested and produced expected results. Based on my own evaluation of the manuscript, no extra experiments or analysis is necessary.

Minor fixes:

Supplementary S10 is currently missing figure legend.

In the README of the code repo, please add a note of the python version (tested on 3.11) to avoid the following error to occur on environment with python 3.12 and above during installation. `AttributeError: module 'pkgutil' has no attribute 'ImpImporter'`.

In the README of the code repo, please add 'setuptools' as a required package to avoid the following error to occur while executing the software: `ImportError: cannot import name 'packaging' from 'pkg_resources'`.

Reviewer #2 (Remarks on code availability):

In this revised version of the manuscript and code repo, the author provided detailed information to install, run, and how to reproduce the results in the manuscript. I'm satisfied with their effort to make the paper reproducible and providing useful resource for the community. However, some minor fixes should be added into the README file.

The installation instruction should include a note that the installation should be done in an environment with python 3.11 (successfully tested on) or lower, higher version of python causes the following error to occur while running `"pip install -e ."`:

`AttributeError: module 'pkgutil' has no attribute 'ImpImporter'`.

After installation following the instructions, I found that setuptools is also a required package to run the software, otherwise the following error would occur:

`ImportError: cannot import name 'packaging' from 'pkg_resources'`

After the above fixes, the software successfully installed and tested on a Linux (x86_64) system with

the provided testing data. The software generates expected output without a problem.

Reviewer #3 (Remarks to the Author):

The authors have addressed my previous concerns.

The ML is more understandable now, including the "hierarchical loss" (which is actually pretty straight forward).

In the newer binning-based analyses, The use of kraken2 and majority voting for classifying MAGs is not ideal (GTDB-Tk is the standard for this), but it is not a major issue.

REVIEWERS' COMMENTS

Reviewer #2 (Remarks to the Author):

The revised version of the manuscript adds extra datasets, tools to compare, and different parameter settings into the benchmarking analysis, which strength to significance and credibility of the manuscript. Adding synthetic datasets that are widely used for benchmarking studies into the analysis fills the gap between pure simulation and real world metagenomic data. The authors also added multiple methods that leverage the usage of abundance vector such as VAMB binner and MetaMaps, and the results suggests that Taxometer performs better overall at species rank annotations.

In addition, the authors replaced Taxometer F1-score with Taxometer precision to evaluate whether the software can be used as a benchmarking tool for the other classifiers. The idea behind using Taxometer as a benchmark tool is to measure the consistency between labels of contigs originated the same taxon, which is sound to me.

The authors also provided a detailed explanation of the algorithm and architecture of the machine learning model used in this method, which makes the method more clear and easier to understand.

Overall, I'm satisfied with the revision of the manuscript. The work supports the conclusions and claims. The software was tested and produced expected results. Based on my own evaluation of the manuscript, no extra experiments or analysis is necessary.

Answer: we thank the reviewer for their effort and insightful comments that greatly helped to improve the article quality.

Minor fixes:

Supplementary S10 is currently missing figure legend.

Answer: we thank the reviewer for pointing that out, the legend issue is now fixed.

In the README of the code repo, please add a note of the python version (tested on 3.11) to avoid the following error to occur on environment with python 3.12 and above during installation. `AttributeError: module 'pkgutil' has no attribute 'ImpImporter'`.

In the README of the code repo, please add 'setuptools' as a required package to avoid the following error to occur while executing the software: `ImportError: cannot import name 'packaging' from 'pkg_resources'`.

Reviewer #2 (Remarks on code availability):

In this revised version of the manuscript and code repo, the author provided detailed information to install, run, and how to reproduce the results in the manuscript. I'm satisfied with their effort to make the paper reproducible and providing useful resource for the community. However, some minor fixes should be added into the README file.

The installation instruction should include a note that the installation should be done in an environment with python 3.11 (successfully tested on) or lower, higher version of python causes the following error to occur while running "`pip install -e .`":

`AttributeError: module 'pkgutil' has no attribute 'ImpImporter'`.

After installation following the instructions, I found that setuptools is also a required package to run the software, otherwise the following error would occur:

```
ImportError: cannot import name 'packaging' from 'pkg_resources'
```

After the above fixes, the software successfully installed and tested on a Linux (x86_64) system with the provided testing data. The software generates expected output without a problem.

Answer: we greatly appreciate that the reviewer checked the code with different Python versions and suggested the specific fix to the problem. The issue is now fixed and the updated code made available to the users.

Reviewer #3 (Remarks to the Author):

The authors have addressed my previous concerns.

The ML is more understandable now, including the "hierarchical loss" (which is actually pretty straight forward).

Answer: we thank the reviewer for their time and effort in bringing the need for clarification to our attention.

In the newer binning-based analyses, The use of kraken2 and majority voting for classifying MAGs is not ideal (GTDB-Tk is the standard for this), but it is not a major issue.

Answer: we agree that GTDB-tk is a standard tool for classifying MAGs. We however chose the majority vote strategy to investigate the capacity of VAMB to correct the contigs annotations using only the binning features and the contigs taxonomy, putting it under the same condition as Taxometer operates under. We would also like to point out that this strategy does not require any additional compute, as running GTDB-tk on several thousand bins (a common order of magnitude for many metagenomic datasets) would result in many hours of costly computational jobs, thus not being feasible and reproducible for many of our users.